# ActiveUltraFeedback:
# Efficient Preference Data Generation using Active Learning

**Davit Melikidze** [1] [*]  **Marian Schneider** [1] [*]  **Jessica Lam** [2] [*]  **Martin Wertich** [1] [*]
**Ido Hakimi** [1] [3]  **Barna Pásztor** [1] [3]  **Andreas Krause** [1] [3]

## Abstract

Reinforcement Learning from Human Feedback (RLHF) has become the standard for aligning Large Language Models (LLMs), yet its efficacy is bottlenecked by the high cost of acquiring preference data, especially in low-resource and expert domains. To address this, we introduce ACTIVEULTRAFEEDBACK, a modular active learning pipeline that leverages uncertainty estimates to dynamically identify the most informative responses for annotation. Our pipeline facilitates the systematic evaluation of standard response selection methods alongside DOUBLE REVERSE THOMPSON SAMPLING (DRTS) and DELTAUCB, two novel methods prioritizing response pairs with large predicted quality gaps, leveraging recent results showing that such pairs provide good signals for fine-tuning. Our experiments demonstrate that ACTIVEULTRAFEEDBACK yields high-quality datasets that lead to significant improvements in downstream performance, notably achieving comparable or superior results with as little as one-sixth of the annotated data relative to static baselines. Our pipeline is available at https://github.com/lasgroup/ActiveUltraFeedback and our preference datasets at https://huggingface.co/ActiveUltraFeedback.

## 1. Introduction

Reinforcement Learning from Human Feedback (RLHF) has established itself as a critical methodology to align

---

[*]Equal contribution [1]ETH Zurich [2]Institute of Neuroinformatics, University of Zurich and ETH Zurich [3]ETH AI Center. Correspondence to: Davit Melikidze <dmelikidze@ethz.ch>, Marian Schneider <smarian@ethz.ch>, Jessica Lam <jehong@ethz.ch>, Martin Wertich <mwertich@ethz.ch>.

*Proceedings of the $43^{rd}$ International Conference on Machine Learning*, Seoul, South Korea. PMLR 306, 2026. Copyright 2026 by the author(s).

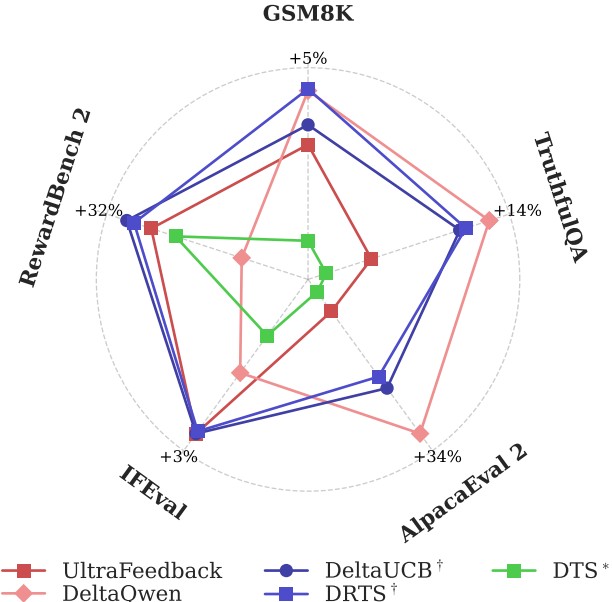

*Figure 1.* Comparison of response pair selection methods on downstream and reward model benchmarks deployed in ACTIVEULTRAFEEDBACK. The scores have been averaged over four datasets (see Section 5.4) of different scales, and indicate improvement over the base model. * denotes an existing dueling bandit method and † indicates our novel active delta learning methods.

Large Language Models (LLMs) with human preferences (Ziegler et al., 2019; Ouyang et al., 2022). RLHF guides the model using human feedback articulated as pairwise preferences over potential outputs, resulting in more naturalistic and human-like behaviour (Christiano et al., 2017). The standard implementation involves training a reward model, followed by model optimization with Proximal Policy Optimization (PPO) (Schulman et al., 2017) to maximize expected rewards (Ouyang et al., 2022). Alternatively, Direct Preference Optimization (DPO) (Rafailov et al., 2023) circumvents the need for a separate reward model by optimizing the model directly on the dataset of pairwise preferences. The potential efficacy of these methods increases with the quality of the preference data, but human annotation is expensive to obtain, especially in low-resource or expert domains. Consequently, a promis-

ing direction for low-cost and scalable preference dataset creation is to reduce annotation requirements by identifying and labelling only the most informative response pairs.

Existing works such as UltraFeedback (Cui et al., 2024), Magpie (Xu et al., 2025), and Nectar (Zhu et al., 2024) generate response pairs through static, passive heuristics. Common choices are random or best-of-$N$ sampling (Cui et al., 2024; Zhu et al., 2024), which are either inefficient or require multiple annotations per prompt. Our experiments show that neither results in high-quality datasets. More recently, the Delta Learning Hypothesis (DLH) (Geng et al., 2025) proposed a novel approach by pairing models of different sizes within a single family (e.g., small vs. large) to form contrastive pairs without annotation. While effective for common applications, this rigidity limits DLH to domains within the chosen model family's training data, and as our experiments show, its performance is limited to DPO fine-tuning. Therefore, the question of collecting high-quality preference datasets not tied to specific algorithms while keeping the need for costly annotation low remains open.

In this work, we propose ACTIVEULTRAFEEDBACK, a modular preference data collection pipeline. Our framework is modeled after the contextual dueling bandit problem (Dudík et al., 2015). In this setup, the prompt serves as the context, and the objective is to select two candidate responses (the arms) from a diverse pool for annotation. We maintain a probabilistic estimate of response quality, updated sequentially as data is collected, to guide the selection of subsequent pairs. Within this framework, we conduct a systematic evaluation of response pair selection methods, comparing standard dueling bandit approaches against established heuristics. Furthermore, we introduce DOUBLE REVERSE THOMPSON SAMPLING (DRTS) and DELTAUCB, two novel methods integrating the insights of the Delta Learning Hypothesis (Geng et al., 2025) by prioritizing pairs with high predicted quality gaps rather than simply minimizing regret. As previewed in Figure 1, ACTIVEULTRAFEEDBACK with DRTS and DELTAUCB consistently outperforms prior heuristics and standard dueling-bandit baselines across both fine-tuned and reward-model benchmarks. Notably, ACTIVEULTRAFEEDBACK demonstrates strong sample-efficiency, matching or outperforming previous methods using only one-sixth of the data, requiring only a single pairwise comparison per prompt for annotation, and not being confined to a single model family. This efficiency enables its application to domains not supported by previous methods. Our detailed ablations demonstrate that these results hold across various datasets and fine-tuning algorithms.

In summary, our contributions are as follows:

- We introduce ACTIVEULTRAFEEDBACK, a modular preference data generation pipeline, that can be deployed with any response selection and uncertainty

quantification methods to guide data collection.

- We are the first to perform a systematic comparison of dueling bandit acquisition functions and common data collection heuristics across a comprehensive evaluation suite covering both reward modeling and diverse downstream benchmarks.

- We introduce two novel response pair selection approaches, DRTS and DELTAUCB, that generate datasets yielding strong performance across prompt sources, tasks, and fine-tuning algorithms, while relying on fewer annotations.

- We open-source ACTIVEULTRAFEEDBACK to allow for easy adoption in existing data pipelines and release artifacts, such as datasets and models.

## 2. Related Work

Reinforcement Learning from Human Feedback (RLHF) is a common method for training models on qualitative objectives concerning human preferences (Christiano et al., 2017; Ziegler et al., 2019; Ouyang et al., 2022). A standard pipeline involves training a reward model on pairwise comparison data, then reinforcement learning algorithms like PPO (Schulman et al., 2017) optimize the model. Alternatively, Direct Preference Optimization (DPO) (Rafailov et al., 2023) offers a solution that combines the two steps. However, the efficacy of these methods is bottlenecked by data provenance. Traditional pipelines rely on manual annotation (Ziegler et al., 2019; Stiennon et al., 2020; Bai et al., 2022) or noisy indirect signals (Ethayarajh et al., 2022). The former is prohibitively expensive to scale, while the latter lacks control over domain coverage and data quality.

To scale up supervision and leverage the performance of frontier models, recent efforts, such as UltraFeedback (Cui et al., 2024), Magpie (Xu et al., 2025), and Nectar (Zhu et al., 2024) have shifted towards generating synthetic data. They follow a common paradigm: a pool of instruction-tuned LLMs generates multiple candidate responses per prompt, then the candidates are scored or ranked (Zhu et al., 2024) by a judge, and a chosen-rejected pair is selected (Cui et al., 2024; Wang et al., 2024a). While these methods have successfully trained open-source models like Zephyr (Tunstall et al., 2024), Tulu 3 (Lambert et al., 2025), and Olmo 2 (Walsh et al., 2025), they apply the same selection strategy to every prompt regardless of response quality uncertainty. This lack of adaptivity often results in sample inefficiency and low-quality datasets, as the system consumes budget on trivial comparisons while missing high-information pairs. Alternatively, the Delta Learning Hypothesis (DLH) (Geng et al., 2025) employs a structural heuristic, pairing models of different sizes (e.g., 0.6B vs. 32B) within a single family to guarantee a quality gap without requiring a

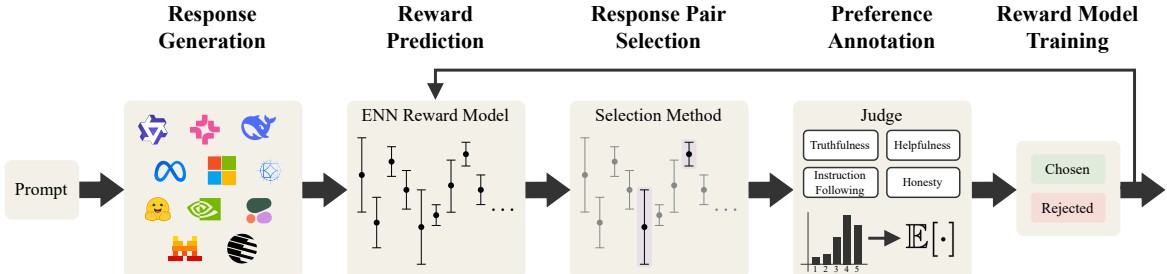

*Figure 2.* The ACTIVEULTRAFEEDBACK pipeline. For each prompt, responses are generated from a large pool of LLMs, the rewards for the responses are predicted with corresponding uncertainties, and a pair of responses is selected for preference annotation. Each new batch of preference data is used to train the reward model, improving the accuracy of reward and uncertainty estimates for subsequent iterations. The displayed procedure is performed in a looping manner until all prompts have been processed.

judge. Despite its success in training Olmo 3 (Olmo et al., 2025) and SmolLM3 (Bakouch et al., 2025), DLH is rigidly confined to intra-family comparisons, limiting its applicability to their often unknown training domains.

Recent works address sample inefficiency in RLHF by formulating it as a contextual duelling bandit problem (Dudik et al., 2011). For reward model training, prior work adapts Double Thompson Sampling (DTS) (Dwaracherla et al., 2024), applies information-theoretic selection (Shen et al., 2025), and uses uncertainty to estimate preference quality and adaptively weight samples (Zhang et al., 2025). For model fine-tuning, uncertainty estimates over predicted rewards improve sample efficiency through uncertainty-based data selection (Liu et al., 2024c; Muldrew et al., 2024; Mehta et al., 2025; Cercola et al., 2026), exploration bonuses (Liang et al., 2022), or uncertainty-regularized objectives that penalize high-uncertainty rewards during RL optimization (Zhai et al., 2026). However, the literature remains fragmented: studies typically focus narrowly on either reward model training (Dwaracherla et al., 2024; Shen et al., 2025; Zhang et al., 2025) or policy optimization (Muldrew et al., 2024; Liu et al., 2024c; Kveton et al., 2025; Mehta et al., 2025), often within a single model family. In contrast, we do not restrict our scope to a single selection method, application, or optimization algorithm.

We bridge this gap by proposing a unified, modular pipeline that enables evaluating response pair selection strategies across both downstream fine-tuning and reward modeling. Within this framework, we benchmark active learning strategies directly against static heuristics and introduce novel methods that operationalize insights from the Delta Learning Hypothesis. Our pipeline generates high-quality datasets for both reward modeling and model fine-tuning, and performs well with multiple preference optimization algorithms.

## 3. Background

Reinforcement Learning from Human Feedback (RLHF) aligns models with human intent by learning from a dataset of pairwise comparisons $\mathcal{D} = \{(x_i, y_i^+, y_i^-)\}_{i=1}^N$, where $x_i$ denotes a prompt and $(y_i^+, y_i^-)$ denotes candidate responses with $y_i^+$ preferred to $y_i^-$. For brevity, we drop the indexing by $i$ for this section. The standard approach (Christiano et al., 2017) proceeds in two stages. First, a reward model $r_\phi(x, y)$ is trained to approximate the latent human preference distribution. This typically relies on the Bradley-Terry model (Bradley & Terry, 1952), which assumes that the comparison feedback is drawn from a Bernoulli distribution and the probability of $y^+$ being preferred to $y^-$ is given by the sigmoid of their reward difference, i.e.,

$$p(y^+ \succ y^- \mid x) = \mathrm{s}(r(x, y^+) - r(x, y^-)), \quad (1)$$

where $\mathrm{s}(x) = (1 + e^{-x})^{-1}$ is the sigmoid function and $r$ is an unknown latent scalar function. The parametrized reward model $r_\phi$ is then optimized to estimate the unknown reward function $r$ by minimizing the negative log-likelihood of the dataset in $\mathcal{D}$. Second, the model, $\pi_\theta$, is optimized to maximize the regularized objective

$$\mathcal{J}(\theta) = \mathbb{E}_{x \sim \mathcal{D}, y \sim \pi_\theta(\cdot|x)}\left[r_\phi(x, y) - \lambda \, \mathrm{KL}(\pi_\theta \| \pi_{\mathrm{ref}})\right], \quad (2)$$

where KL denotes the Kullback-Leibler divergence from a reference model $\pi_{\mathrm{ref}}$ and $\lambda$ controls the strength of the regularization. Direct Preference Optimization (DPO) (Rafailov et al., 2023) is a widely used alternative that improves computational efficiency by combining the reward modeling and policy fine-tuning steps, turning RLHF into a supervised learning task. Regardless of the optimisation approach, standard RLHF methods consider $\mathcal{D}$ as a fixed, static artifact.

While the standard RLHF approaches only use a pointwise estimate for the reward function $r_\phi$, we leverage uncertainty estimates to guide data collection. Let $\underline{r}_\phi(x, y)$ and $\overline{r}_\phi(x, y)$ denote the lower and upper confidence bounds of the reward estimate. Under the Bradley-Terry assumption, the upper confidence bound (UCB) probability $\overline{p}$ that a response $y_j$ is preferred over another response $y_{j'}$ is defined as

$$\overline{p}_\phi(y_j \succ y_{j'}) = \mathrm{s}(\overline{r}_\phi(x, y_j) - \underline{r}_\phi(x, y_{j'})). \quad (3)$$

Conversely, the lower confidence bound (LCB) probability $\underline{p}$ is defined by the worst-case reward difference

$$\underline{p}_\phi(y_j \succ y_{j'}) = \mathrm{s}(\underline{r}_\phi(x, y_j) - \overline{r}_\phi(x, y_{j'})). \qquad (4)$$

These probabilistic bounds serve as the foundation for response selection methods described in Section 4.3.

# 4. The ACTIVEULTRAFEEDBACK Pipeline

In this section, we introduce ACTIVEULTRAFEEDBACK, our scalable and modular pipeline for creating high-quality preference datasets without extensive annotation requirements. Given a set of $N$ prompts, $\mathcal{P} = \{x_i\}_{i=1}^N$, AC-TIVEULTRAFEEDBACK starts with an empty dataset $\mathcal{D} = \emptyset$, processes the prompts in $\mathcal{P}$ iteratively in batches, and appends the new data points to $\mathcal{D}$. Unlike prior work, we present a unified active learning pipeline for preference data generation. ACTIVEULTRAFEEDBACK supports plug-and-play uncertainty-aware acquisition functions, evaluates them across reward modeling and downstream preference tuning, and introduces two new delta-oriented methods, DRTS and DELTAUCB. The five key steps for each batch, illustrated in Figure 2, are as follows:

1. **Response Generation**: For each prompt $x_i$ in the batch, generate a diverse set of candidate responses $\{y_{i,j}\}_{j=1}^m$ from a pool of $m$ LLMs (Section 4.1).

2. **Reward Prediction**: For each prompt–response pair $(x_i, y_{i,j})$, estimate $\underline{r}_\phi(x_i, y_{i,j})$ and $\overline{r}_\phi(x_i, y_{i,j})$ (Section 4.2).

3. **Response Pair Selection**: Select two responses $(y_{i,j}, y_{i,j'})$ for each prompt in the batch for pairwise comparison (Section 4.3).

4. **Preference Annotation**: Collect preference annotations and append the resulting triplets, $(x_i, y_i^+, y_i^-)$, to $\mathcal{D}$ (Section 4.4).

5. **Reward Model Training**: Update the reward model's parameters, $\phi$, with the dataset $\mathcal{D}$ collected thus far (Section 4.5).

## 4.1. Response Generation

Given an input prompt $x_i$, we employ a model pool of $m$ LLMs to generate candidate responses $\{y_{i,j}\}_{j=1}^m$. Our model pool comprises $m = 30$ open-weight LLMs from 12 families, including Qwen 2.5 (Qwen et al., 2025), Qwen 3 (Yang et al., 2025), Llama 3 (Grattafiori et al., 2024), Gemma 3 (Team et al., 2024), and SmolLM 2 (Allal et al., 2025). Following the UltraFeedback pipeline's approach (Cui et al., 2024; Lambert et al., 2025; Walsh et al.,

2025), for each prompt–LLM pair, we select a guiding principle (from "helpfulness", "truthfulness", and "honesty") at random to create more diverse responses.

The combination of aspects and the diverse model pool ensures that the candidate responses provide a broad content and quality diversity for the response pair selection methods. We defer further details on the model pool (Table 3), principles (Section B.2), and the used prompt templates (Section H.1) to the Appendix.

## 4.2. Reward Prediction

To operationalize the uncertainty estimates defined in Section 3, we employ the Epistemic Neural Network (ENN) framework (Osband et al., 2023). Following prior works for active learning in RLHF (Dwaracherla et al., 2024; Melo et al., 2024; Liu et al., 2024c), we implement the ENN as an ensemble of shallow Multi-Layer Perceptrons with a shared, frozen backbone, deriving the final reward $r_\phi(x_i, y_j)$ as the ensemble mean and uncertainty $\sigma_\phi(x_i, y_j)$ as the standard deviation. These quantities define the upper and lower confidence bounds for the reward estimate

$$\overline{r}_\phi(x_i, y_j) = r_\phi(x_i, y_j) + \beta\sigma_\phi(x_i, y_j),$$
$$\underline{r}_\phi(x_i, y_j) = r_\phi(x_i, y_j) - \beta\sigma_\phi(x_i, y_j)$$

respectively, where $\beta > 0$ is a scaling parameter, as well as the UCB $\overline{p}_\phi$ (Equation (3)) and LCB $\underline{p}_\phi$ (Equation (4)) for comparisons between response pairs. Additional details on the network architecture are provided in Section C.1.

## 4.3. Response Pair Selection

For each prompt $x_i$, we select a response pair $(y_{i,j}, y_{i,j'})$ for preference annotation using a response pair selection method. We explore four baseline heuristics that do not make use of the reward estimates and three methods developed for the Dueling Bandit problem (Bengs et al., 2021). Additionally, we propose two novel methods, DRTS and DELTAUCB, based on the Delta Learning Hypothesis (DLH) (Geng et al., 2025). We provide an overview of the algorithms here and defer further details to Section D.

**Baseline Heuristics** We evaluate four passive baseline heuristics that operate independently of reward estimates. (i) RANDOM samples a pair uniformly at random from the candidate set; (ii) MAXMIN queries a judge for the entire candidate set to identify the responses with the highest and lowest quality; (iii) ULTRAFEEDBACK (Cui et al., 2024) samples four responses uniformly at random, queries a judge on their quality, and returns the highest-scoring one as the preferred response paired with a randomly selected one from the remaining three; (iv) DELTAQWEN (Geng et al., 2025) selects the responses generated by the Qwen 3 0.6B and 32B models, with the latter considered as the preferred response.

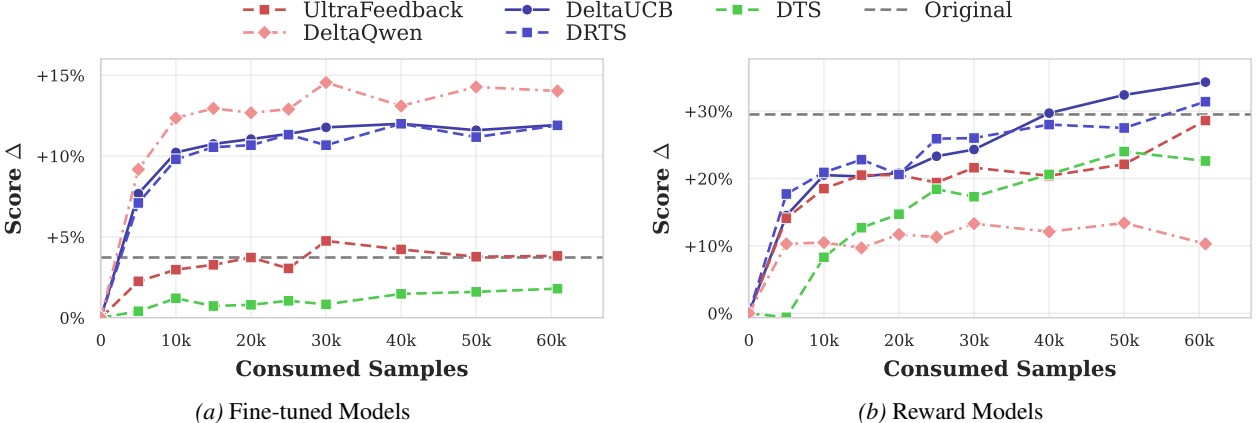

*Figure 3.* Mean performance trajectories for fine-tuned and reward models as a function of consumed samples on the ACTIVEULTRA-FEEDBACK prompt pool using DPO. All curves share the same prompts and differ only in the response pair selection strategy. For readability, we subset the selection strategies; for a full comparison, see Figure 8 in the Appendix. We compare datasets generated via ACTIVEULTRAFEEDBACK using various selection methods, and also report the score achieved by the original **UltraFeedback** dataset (Cui et al., 2024) with its original response pairs.

*Table 1.* Overview of response pair selection methods and the number of responses that need to be annotated per prompt. † indicates the methods that we propose.

| Methods | # Responses to Annotate |
|---|---|
| *Baseline Heuristics* | |
| RANDOM | 2 |
| MAXMIN | $m$ |
| ULTRAFEEDBACK (Cui et al., 2024) | 4 |
| DELTAQWEN (Geng et al., 2025) | 0 |
| *Dueling Bandit Methods* | |
| INFOMAX (Saha, 2021) | 2 |
| DTS (Wu & Liu, 2016) | 2 |
| MAXMINLCB (Pásztor et al., 2024) | 2 |
| *Active Delta Learning Methods* | |
| DRTS† | 2 |
| DELTAUCB† | 2 |

**Dueling Bandit Methods** We adopt three acquisition functions from prior literature on dueling bandits: (i) IN-FOMAX (Saha, 2021) prioritizes pure exploration by selecting the response pair with the highest joint uncertainty, regardless of the predicted reward quality: $\arg\max_{j\neq j'}\overline{p}_\phi(y_{i,j}\succ y_{i,j'})-\underline{p}_\phi(y_{i,j}\succ y_{i,j'})$; (ii) DOU-BLE THOMPSON SAMPLING (DTS) (Wu & Liu, 2016) addresses the exploration-exploitation trade-off by drawing two independent samples from the reward posterior and selecting the responses that maximize them; (iii) MAXMINLCB (Pásztor et al., 2024) considers the pairwise LCB (Equation (4)) and selects the pair $(j_1, j_2)$ where $j_1 = \arg\max_j \min_{j'\neq j}\underline{p}_\phi(y_j\succ y_{j'})$ maxi-

mizes the minimum LCB against any other response, and $j_2 = \arg\min_{j\neq j_1}\underline{p}_\phi(y_{j_1}\succ y_j)$ minimizes the LCB against $j_1$. These algorithms offer no-regret guarantees (DTS, MAXMINLCB) or sample complexity bounds for identifying the optimal response (INFOMAX).

**Active Delta Learning Methods** We introduce two novel methods based on the Delta Learning Hypothesis (Geng et al., 2025), which states that the absolute quality of the responses is less important than the relative difference, and proposed the DELTAQWEN method introduced above.

DOUBLE REVERSED THOMPSON SAMPLING (DRTS) selects one response that maximizes and another that minimizes their respective samples from the reward posterior. This strategy explicitly targets pairs with a significant delta in quality, while the underlying stochastic sampling preserves exploration and diversity.

DELTAUCB identifies pairs with the largest optimistic quality difference by selecting the pair $(y_{i,j}, y_{i,j'})$ that maximizes the probability that $j$ is preferred over $j'$ in the best-case scenario: $\arg\max_{j\neq j'}\overline{p}_\phi(y_{i,j}\succ y_{i,j'})$. By relying on these optimistic bounds, DELTAUCB guides exploration toward pairs that plausibly exhibit significant quality differences, without requiring stochastic sampling.

### 4.4. Preference Annotation

After the response pairs $(y_{i,j}, y_{i,j'})$ for each prompt $x_i$ are selected, we query a judge for the pairwise comparison feedback and, following the annotation, append $(x_i, y_i^+, y_i^-)$ to the dataset $\mathcal{D}$. To facilitate scalable and reproducible experiments, we employ a large LLM instead of human annotators. Our goal is not to fully replace human preference

*Table 2.* Comparison between all response pair selection methods, based on the reward model and fine-tuned model (DPO) performance after training the same base model on each generated dataset. The base model score is given for reference, and all scores are reported as relative deltas to it, with higher values indicating better performance. We also provide the deltas achieved with the original response pairs in **UltraFeedback**. † denotes our proposed methods. Best scores are marked in bold.

| Method | GSM8K | IFEval | TruthfulQA | AlpacaEval 2 | Mean | RewardBench 2 |
|---|---|---|---|---|---|---|
| Base Model | 0.758 | 0.713 | 0.468 | 0.083 | 0.506 | 0.290 |
| Original | +0.039 | +0.025 | +0.055 | +0.030 | +0.037 | +0.295 |
| RANDOM | +0.024 | +0.028 | +0.056 | +0.077 | +0.046 | +0.278 |
| ULTRAFEEDBACK | +0.037 | -0.001 | +0.039 | +0.072 | +0.036 | +0.287 |
| MAXMIN | +0.022 | -0.016 | **+0.150** | +0.289 | +0.111 | +0.318 |
| DELTAQWEN | **+0.055** | +0.047 | +0.130 | **+0.316** | **+0.137** | +0.100 |
| INFOMAX | +0.011 | +0.019 | +0.018 | +0.020 | +0.016 | +0.297 |
| DTS | +0.011 | +0.034 | +0.013 | +0.037 | +0.023 | +0.224 |
| MAXMINLCB | +0.015 | +0.017 | +0.006 | +0.027 | +0.016 | +0.230 |
| DRTS† | **+0.055** | **+0.050** | +0.143 | +0.259 | +0.127 | +0.312 |
| DELTAUCB† | +0.040 | +0.025 | +0.137 | +0.281 | +0.120 | **+0.339** |

judgments, but to enable controlled large-scale comparisons of response pair selection methods while reducing annotation costs. Specifically, a judge LLM independently scores each response on a 1–5 Likert scale across four quality aspects: truthfulness, instruction following, honesty, and helpfulness. The response with the highest average score is then selected as preferred. To ensure high-quality labels, we validated our annotation setup through extensive experiments comparing different judges, prompting strategies, and scoring mechanisms. Further details are provided in Section E.

### 4.5. Reward Model Training

Finally, we update the ENN model to improve its reward estimates using the latest batch of preference data combined with previously collected samples. For details on hyperparameters and the training procedure, see Section C.2.

## 5. Evaluation

In this section, we evaluate the response pair selection methods (Section 4.3) deployed in ACTIVEULTRAFEEDBACK by investigating the following research questions:

1. **Performance**: Can ACTIVEULTRAFEEDBACK generate high-quality datasets (Section 5.2), and which response pair selection method achieves the best performance?

2. **Efficiency**: Does active response pair selection provide sample efficiency improvements (Section 5.3), yielding equal or higher scores using fewer annotated samples?

3. **Generalization**: Do results generalize across prompt

datasets (Section 5.4) and preference optimization algorithms (Section 5.5)?

### 5.1. Implementation Details

**Datasets** We choose the **UltraFeedback** dataset[1] (Cui et al., 2024) as our primary set of prompts $\mathcal{P}$ and consider further prompt collections in Section 5.4.

**Evaluation** To evaluate the datasets collected by ACTIVEULTRAFEEDBACK, we consider the two steps of RLHF described in Section 3, reward model training and model fine-tuning, separately. First, we train a standard reward model using the negative log likelihood minimization of the Bradley-Terry model defined in Equation (1) and evaluate it on the RewardBench 2 benchmark (Malik et al., 2025). To keep our evaluation protocol standardized, we train the reward model independently of the ENN described in Section 4.2. To isolate reward modeling and preference fine-tuning, we use DPO (Rafailov et al., 2023), which combines the two steps of RLHF. We evaluate other direct optimization algorithms in Section 5.5. The fine-tuned models are then evaluated on the GSM8K (Cobbe et al., 2021), IFEval (Zhou et al., 2023), TruthfulQA (Lin et al., 2022), and AlpacaEval 2 (Dubois et al., 2024) benchmarks covering the crucial capabilities of mathematical reasoning, instruction-following, knowledge recall, and human preference. Both trainings for evaluation are initialized from the Tulu 3 8B SFT model[2] (Lambert et al., 2025) and all scores are reported as deltas relative to the base model. We measured our results' sensitivity to the inherent stochastic nature of our pipeline and consider a difference of at least 0.008 for

---

[1] allenai/ultrafeedback_binarized_cleaned
[2] allenai/Llama-3.1-Tulu-3-8B-SFT

the downstream benchmarks and 0.02 for RewardBench 2 to be significant. Detailed analysis is provided in Section F.2. We carry out hyperparameter tuning for both the response pair selection methods from Section 4.3 and the training methods used for evaluation. Further implementation details are provided in Section F.

## 5.2. Response Pair Selection Methods

In this section, we address our first research question by employing the ACTIVEULTRAFEEDBACK pipeline with the response pair selection methods described in Section 4.3. The results presented in Table 2 and Figure 1 demonstrate that ACTIVEULTRAFEEDBACK with DRTS and DELTAUCB can generate high-quality datasets for both reward modeling and preference optimization, outperforming all other methods except DELTAQWEN for the latter. This is expected due to the known performance of DELTAQWEN for fine-tuning with DPO on common domains and datasets. However, it significantly lags behind even random sampling for reward modelling. We attribute this discrepancy for DELTAQWEN to its confinement to the training distribution of the underlying models.

Contrary to many prior works considering active learning for RLHF as a contextual dueling bandit problem (Section 2), we find that previously proposed dueling bandit methods do not transfer effectively to the task of preference data generation. Analyzing the generated datasets (Section G.1) confirms that DTS and MAXMINLCB successfully achieve their theoretical goal of identifying high-quality responses, but yield datasets that lack the quality deltas required for learning. Consequently, these methods underperform even random sampling, demonstrating that the objectives of regret minimization and uncertainty minimization are misaligned with the goal of preference data generation. Intuitively, delta-based selection improves dataset quality by creating clearer preference boundaries. Larger quality gaps provide a lower-noise training signal than ambiguous comparisons between similarly strong answers. Appendix Section G.1 supports this interpretation: DRTS and DELTAUCB retain high chosen-response scores while pairing them with substantially weaker rejected responses, whereas regret-minimizing methods such as DTS and MAXMINLCB often compare two strong answers and thus provide weaker supervision.

## 5.3. Sample Efficiency

We address our second research question by evaluating models trained on subsets of the generated datasets. The results for downstream benchmarks (Figure 3a) show that our proposed methods, DRTS and DELTAUCB, demonstrate strong sample-efficiency in downstream evaluations. Using our proposed methods, models fine-tuned on merely 5'000 to 10'000 samples outperform those trained on 60'000 samples

from the datasets generated using RANDOM, ULTRAFEED-BACK, or dueling bandit methods. Notably, they also lead to better performance than when training on the original **Ultra-Feedback** dataset (Cui et al., 2024). While DELTAQWEN shows a 1% improvement in mean downstream score over DRTS, this is driven disproportionately by AlpacaEval 2 performance, as also shown on Table 2 (see Appendix, Figure 7). Notably, DELTAUCB shows smaller fluctuations in performance than MAXMIN, DELTAQWEN, and DRTS. These results indicate that DPO training can be made significantly more sample-efficient than previously reported by leveraging optimal selection of responses, and that training models on preference feedback could be achieved at a much lower annotation cost. Our gains are primarily in *annotation efficiency*, not in minimizing raw response-generation cost. In the current implementation, we pre-computed responses and judge scores for the full prompt set to support large-scale ablations, increasing upfront compute relative to static pipelines. However, strong selection methods such as DRTS and DELTAUCB achieve similar downstream performance with substantially fewer selected pairs, reducing end-to-end data collection when annotation is the main bottleneck. Raw compute estimates are given in Section F.4.

As shown on Figure 3b, reward modeling follows a more gradual saturation curve, requiring 40'000 samples to attain benchmark scores equivalent to training on the complete dataset without active response pair selection. Furthermore, Figure 3 reveals a critical limitation of the DELTAQWEN baseline: its strong downstream performance (Figure 3a) contrasts with poor generalization in reward modeling (Figure 3b). In addition, RANDOM shows strong performance for reward modeling, which, in turn, suggests that diversity is a more desirable property for this task than qualitative difference. On the contrary, DRTS and DELTAUCB not only achieve high scores on both tasks but only these two methods are both practical and yield datasets that can surpass the quality of the original one.

## 5.4. Input Prompt Dataset Ablation

To assess the generalization capabilities of ACTIVEULTRA-FEEDBACK beyond the **UltraFeedback** prompts, we evaluate the pipeline on three additional datasets of varying scales: (i) **Skywork** Reward Preference 80k v0.2[3] (Liu et al., 2024b), a high-quality dataset of 80'000 prompts for reward modeling; (ii) **Combined**: a combination of the **UltraFeedback** and **Skywork** datasets with 140'000 prompts; and (iii) **Tulu 3** 8B Preference Mixture[4], a dataset of 272'000 prompts for LLM fine-tuning (Lambert et al., 2025).

Figure 4 confirms that ACTIVEULTRAFEEDBACK, combined with our DRTS and DELTAUCB methods, general-

---

[3]Skywork/Skywork-Reward-Preference-80K-v0.2
[4]allenai/llama-3.1-tulu-3-8b-preference-mixture

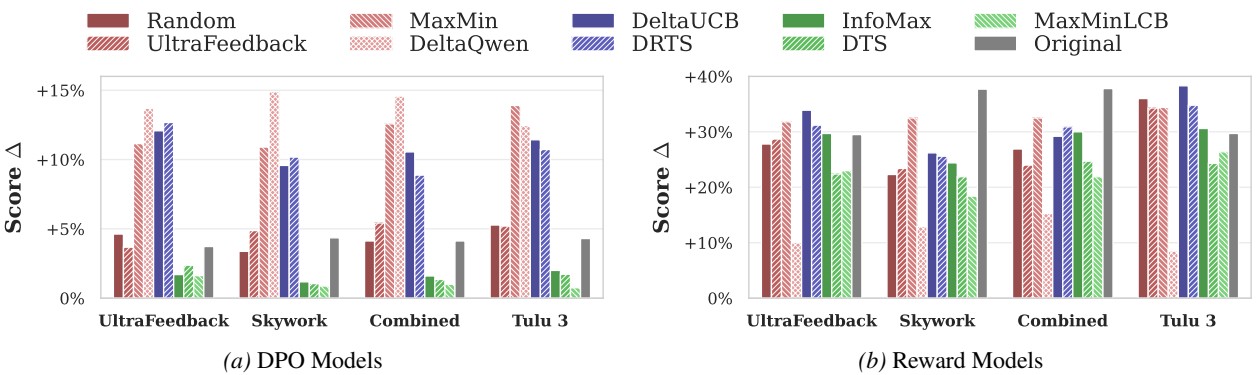

*(a)* DPO Models      *(b)* Reward Models

*Figure 4.* Benchmarking of downstream and reward model performance across input prompt datasets, increasing in scale from left to right. Scores are reported as relative deltas to the base model. We provide the scores achieved using the original preference dataset instead of just the prompts with ACTIVEULTRAFEEDBACK for reference.

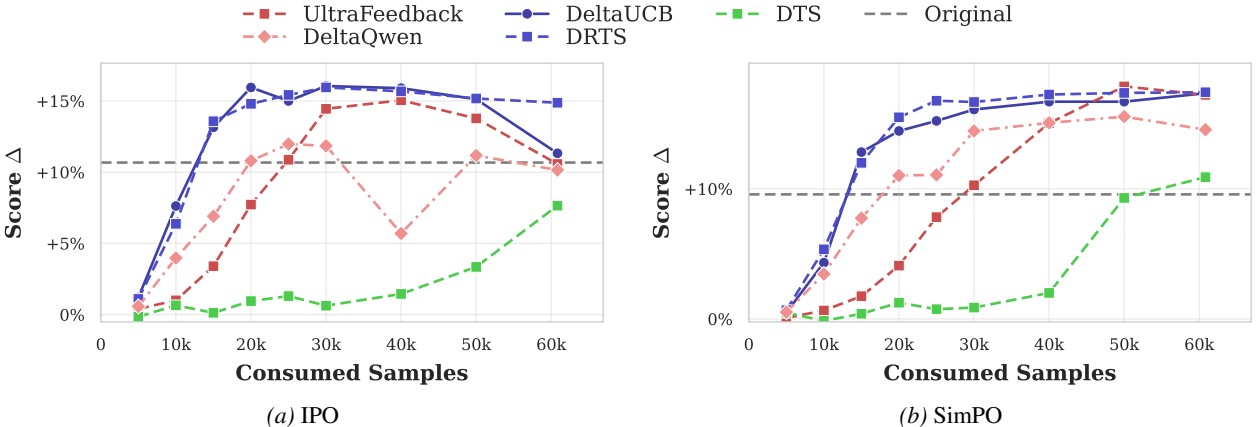

*(a)* IPO      *(b)* SimPO

*Figure 5.* Mean performance trajectories for our fine-tuned models using IPO (Figure 5a) and SimPO (Figure 5b) as a function of consumed samples on datasets generated using ACTIVEULTRAFEEDBACK based on **UltraFeedback** prompts. For readability, we subset the selection strategies, for a full comparison, see Figure 9 in the Appendix. We provide the scores achieved using the original preference dataset instead of just the prompts with ACTIVEULTRAFEEDBACK for reference.

izes effectively across diverse prompt datasets, consistently outperforming existing preference data generation heuristics and standard methods. While DELTAQWEN achieves a high downstream score, similar to Section 5.3, this performance is skewed by AlpacaEval 2 (see Table 24 for exact scores). DELTAQWEN still significantly underperforms on Reward-Bench 2, which we, again, attribute to a lack of diversity.

Remarkably, our pipeline demonstrates substantial improvements over the widely-adopted original preference datasets included in Figure 4 (**UltraFeedback**, **Skywork**, and **Tulu 3**). In terms of DPO mean scores, DRTS and DELTAUCB yield significantly better results across all prompt sources. While the reference **Skywork** and **Combined** datasets retain an advantage in reward model training, which is expected as **Skywork** is curated for reward modelling, our active delta learning methods outperform the baselines on the **Ultra-Feedback** and **Tulu 3** prompts.

## 5.5. Preference Optimization Algorithm Ablation

To evaluate the generalizability of ACTIVEULTRAFEED-BACK across different preference optimization algorithms beyond DPO (Rafailov et al., 2023), we extend our analysis in Section 5.2 to the IPO (Du et al., 2024) and SimPO (Meng et al., 2024) algorithms. While DPO optimizes the policy by implicitly maximizing a reward function with KL-regularization, IPO maximizes the win rate against a fixed policy, eliminating the need for a reward model, and SimPO simplifies the objective by using a length-normalized reward margin for regularization. The results are visualized in Figure 5.

Regardless of the optimization algorithm, DRTS and DELTAUCB remain among the highest performing methods, and their trajectories demonstrate the superior sample efficiency by converging to their top performance using significantly fewer samples than all other methods. In contrast, DELTAQWEN suffers a significant performance drop on

these alternative algorithms, demonstrating its inflexibility and limiting its applicability to very specific experimental setups. We attribute this behavior to the limited diversity of the DELTAQWEN datasets. Unlike DRTS and DELTAUCB, which select from the full model pool, DELTAQWEN always compares the same two Qwen models. This seems sufficient for DPO on common domains, but less robust for IPO and SimPO, which are more sensitive to drift and over-specialization. This is consistent with our diversity analysis in Section G.1.

We observe that RANDOM, ULTRAFEEDBACK, and DTS perform remarkably well with IPO and SimPO, compared to their performance with DPO, but they achieve high performance with large datasets only. Detailed numerical results are provided in Section G.4 and Table 25.

## 6. Conclusion

We present ACTIVEULTRAFEEDBACK, a modular active learning pipeline for preference data generation. ACTIVEULTRAFEEDBACK addresses a central bottleneck in preference optimization: selecting the most informative response pairs for labeling within a limited annotation budget. Our extensive evaluations demonstrate that using datasets produced by ACTIVEULTRAFEEDBACK, particularly when coupled with our novel DRTS and DELTAUCB response selection methods, results in significantly stronger reward and fine-tuned models compared to those derived from static heuristics. Notably, these gains are consistent across varying prompt sources and optimization algorithms, making our approach the first to produce high-quality datasets agnostic to the downstream task or training algorithm. In practical terms, our results suggest three takeaways. First, when annotation budget is limited, DRTS and DELTAUCB are strong default choices. Second, selecting the single best response is not the same as selecting the most informative response pair. Third, while DELTAQWEN is competitive for DPO on common domains, it generalizes less reliably across tasks. Overall, we recommend ACTIVEULTRAFEEDBACK with DRTS or DELTAUCB for efficient preference-data collection.

Importantly, ACTIVEULTRAFEEDBACK is designed as a *platform* for preference-data collection, enabling researchers and practitioners to rapidly develop, swap, and benchmark new methods, uncertainty estimators, and judges. We see many promising directions of future work to build on this platform, such as testing additional uncertainty estimation approaches, setting explicit diversity constraints, incorporating prompt selection into the active learning loop, creating open-source datasets for expert and low-resource domains, and extending the platform with a user interface to collect human annotations. Furthermore, we recognize that the current pipeline incurs substantial computational cost due

to generating responses from many LLMs for each prompt. Therefore, we see strong potential in selecting models to query for responses instead of selecting between already generated responses as a high priority. To lower the barrier to entry and make this line of research more accessible, we therefore release all generated datasets, enabling future researchers to build upon our results without incurring the full computational overhead.

## Acknowledgments

This work was supported as part of the Swiss AI initiative by a grant from the Swiss National Supercomputing Centre (CSCS) on Alps. Barna Pásztor was primarily supported by the ETH AI Center through an ETH AI Center doctoral fellowship, and Ido Hakimi was primarily supported by the ETH AI Center through an ETH AI Center postdoctoral fellowship.

## Impact Statement

This paper presents ACTIVEULTRAFEEDBACK, an active learning pipeline for preference-data collection in RLHF that improves sample efficiency and reduces reliance on human annotation, potentially broadening access to preference optimization and enabling faster iteration on alignment datasets across diverse domains. As with other preference-based approaches, ACTIVEULTRAFEEDBACK may amplify biases in prompts, annotators, or judges, and stronger reward models may increase the risk of reward hacking or over-optimization; while it does not introduce new capabilities for generating harmful content, it could be misused to more efficiently optimize models toward undesirable preferences. We mitigate these risks through evaluation across diverse prompt sources and benchmarks, release of code and datasets for reproducibility and auditing, and a modular design that allows practitioners to incorporate improved judges, safety filters, and bias-mitigation strategies. We encourage future deployments to pair preference-data collection with clear annotation guidelines, safety-focused evaluations, and monitoring for distribution shift and reward-model failures.

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

# Contents of Appendix

# A. Author Contributions

All authors provided valuable contributions to implementing, analyzing, and iterating on experiments, writing and the paper, and managing the overall progress of the project.

**DM** proposed and implemented the DRTS and DeltaUCB algorithms, and conducted extensive experiments evaluating response pair selection methods, generalization to diverse input prompt datasets, and comprehensive ablations on both sample efficiency and preference optimization algorithms. Furthermore, he developed and maintained major parts of the coding infrastructure, including the acquisition functions (such as MaxMinLCB and DTS), the main active learning loop for preference data generation, and the training scripts for DPO, IPO, SimPO and Reward Modeling, along with the evaluation scripts for measuring downstream and reward model performance. To scale the system, he added multi-node support for both training and the active learning loop, and optimized the overall pipeline. Additionally, **DM** experimented on and finalized the LLM-as-a-Judge setup, and conducted rigorous hyperparameter searches to ensure pipeline robustness and maximize performance across all active learning and training components. He led the release of the preference data and trained models and actively contributed to the writing of all sections of the paper.

**MS** developed and maintained major parts of the coding infrastructure, in particular the core inference pipeline used for both response generation and annotation. He curated the model pool, balancing diversity and quality, and built the surrounding infrastructure. Based on this, he conducted experiments and ablations on the model pool and judge. Furthermore, **MS** implemented evaluation pipelines for reward models, downstream benchmarks, and judge models, and contributed to the implementation of acquisition functions from prior work. He contributed to the training and efficient sweeping setup. Building on this, he conducted experiments on the effect of different judges and acquisition functions on reward models and downstream task performance, informing design choices in the final pipeline. Finally, he led the open-source release of the codebase and actively contributed to writing all sections of the paper.

**JL** worked on the response generation stage of the pipeline, contributed to the implementation of acquisition functions from existing works, and proposed the MaxMin baseline. She established schemas to ensure consistency and compliance across the data and methods, and led the open-source release of the dataset. She analysed several experimental design choices, including response quality annotation through probabilistic scoring with LLM-as-a-Judge and the validity of using different judge models for AlpacaEval 2. In addition, **JL** actively contributed to optimization and refactoring efforts, including improvements to the active learning loop, as well as to the writing of all sections of the paper and rebuttal.

**MW** contributed to the implementation and analysis of the active preference data generation pipeline and ran reward-model and downstream evaluations. He was primarily responsible for writing the Abstract, Introduction, Related Work, Conclusion, and the Appendix sections, and contributed to editing the remaining sections of the paper. Moreover, **MW** contributed to the rebuttal and the corresponding refactoring of the paper.

**IH**, **BP**, and **AK** supervised the research, suggested ideas and experiments, guided the project direction, and assisted with writing and editing the paper.

# B. Response Generation

This section details the response generation step (Section 4.1) of ACTIVEULTRAFEEDBACK, in which we use vLLM (Kwon et al., 2023) with a large model pool of diverse LLMs to generate candidate responses for the input prompts.

## B.1. Model Pool

Table 3 lists the 30 LLMs forming our model pool. We include a wide range of both model families (12 different model families, e.g. Qwen 2.5 (Qwen et al., 2025), Qwen 3 (Yang et al., 2025), Llama 3 (Grattafiori et al., 2024), Phi 4 (Abdin et al., 2024), Mistral Large 2 (Mistral AI Team, 2024), Mistral Small 3 (Mistral AI Team, 2025), Nemotron (Wang et al., 2024b; Singhal et al., 2025), Gemma 3 (Team et al., 2024), OLMo 2 (Walsh et al., 2025), Tulu 3 (Lambert et al., 2025), SmolLM 2 (Allal et al., 2025), Moonlight (Liu et al., 2025), Command A (Cohere et al., 2025), and DeepSeek V3 (Liu et al., 2024a)) and model sizes (0.5B to 671B) to ensure content and quality diversity, in line with prior work (Cui et al., 2024; Lambert et al., 2025).

## B.2. Response Principles

Beyond model diversity (Section B.1), we introduce diversity through guiding principles that the LLMs should follow when generating responses. For every prompt-model pair, we uniformly sample a guiding principle among truthfulness, honesty, and helpfulness, at random following the UltraFeedback pipeline's approach (Cui et al., 2024). To demonstrate the principle to the model, we then randomly sample a system prompt, among 11 system prompts, for the sampled principle. We adopt the prompt templates from the UltraFeedback pipeline but explicitly exclude the verbalized calibration principle. This modification prevents the subsequent annotation step (Section E) from being biased by the model's self-expressed uncertainty, which could otherwise lead to artificially lower scores for responses where the model expresses doubt. See Section H.1 for the system prompts.

# C. ENN Reward Model

Following prior work (Dwaracherla et al., 2024; Melo et al., 2024; Liu et al., 2024c), we utilize the Epistemic Neural Network (ENN) (Osband et al., 2023) architecture, implemented by (Yang et al., 2026), to model the reward function. Unlike standard reward models (Section 3) that provide a single scalar point estimate, an ENN represents a distribution over reward functions, $p(r|\mathcal{D})$, where $\mathcal{D}$ is the set of observed preferences. This allows the model to quantify the epistemic uncertainty, the uncertainty stemming from a lack of data, which is the foundation for our active learning methods.

## C.1. Architecture

We implement the ENN using an ensemble built on top of a fixed, pre-trained language model. This architecture consists of two components: a shared backbone and an ensemble of reward heads.

First, for any prompt-response pair $(x, y)$, we extract a feature vector $h(x, y)$ using a pre-trained LLM backbone. We utilize the embedding of the final token from the last hidden layer as the representation. Crucially, this backbone is kept frozen and unchanged during training, so only the lightweight reward heads are updated in the active learning loop, keeping in-loop training efficient and consistent with prior ENN-based RLHF work.

Second, the reward function is estimated by an ensemble of $K$ independent Multi-Layer Perceptrons (MLPs), denoted as $\{r_{\phi_k}\}_{k=1}^{K}$. Each head $k$ takes the embedding $h(x, y)$ as input and outputs a scalar reward. We define the final reward estimate as the mean of the ensemble predictions, $r(x, y)$, while the epistemic uncertainty is quantified by their standard deviation, $\sigma_r(x, y)$. The epistemic uncertainty is scaled by a hyperparameter $\beta > 0$ to obtain the lower and upper bounds of the reward estimate, $\underline{r}(x, y) = r(x, y) - \beta\sigma_r(x, y)$ and $\overline{r}(x, y) = r(x, y) + \beta\sigma_r(x, y)$ respectively. Studying how smaller backbones or ensemble sizes affect downstream acquisition quality is an important direction for future work.

## C.2. Training

We update the ENN reward model at the end of each ACTIVEULTRAFEEDBACK iteration using a replay buffer $\mathcal{B} = \{(x_i, y_i^+, y_i^-)\}$ that aggregates all preference data collected thus far. We sample (without replacement) a training dataset $\mathcal{D}_{\text{train}}$ by sampling from $\mathcal{B}$ such that its size is given by $|\mathcal{D}_{\text{train}}| = \min(|\mathcal{B}|, b \cdot \rho)$, where $b$ denotes the ACTIVEULTRAFEEDBACK batch size and $\rho$ is a hyperparameter controlling the magnitude of $\mathcal{D}_{\text{train}}$.

The parameters $\phi = \{\phi_k\}_{k=1}^{K}$ for the $K$ ensemble heads are updated on $\mathcal{D}_{\text{train}}$ by minimizing the regularized Bradley-Terry negative log-likelihood:

$$\mathcal{J}(\phi) = \frac{1}{K} \sum_{k=1}^{K} \Bigg( \mathbb{E}_{(x,y^+,y^-)\sim\mathcal{D}_{\text{train}}} \left[ -\log \text{s} \left( r_{\phi_k}(x, y^+) - r_{\phi_k}(x, y^-) \right) \right]$$
$$+ \gamma \mathbb{E}_{(x,y^+,y^-)\sim\mathcal{D}_{\text{train}}} \left[ (r_{\phi_k}(x, y^+) + r_{\phi_k}(x, y^-))^2 \right] \tag{5}$$
$$+ \zeta \|\phi_k - \widetilde{\phi}_k\|_2^2 \Bigg),$$

where $\text{s}(x) = (1 + e^{-x})^{-1}$ is the sigmoid function. In addition to the standard Bradley-Terry objective, this objective contains two regularization terms. The first term, controlled by $\gamma$, centers the predicted rewards around zero. Since the Bradley-Terry probability is invariant to additive constants ($\text{s}(a - b) = \text{s}((a + c) - (b + c))$), different heads can arbitrarily drift in absolute value. This term prevents such drift, ensuring that the ensemble variance reflects genuine uncertainty

rather than arbitrary offsets between heads. The second term, controlled by $\zeta$, anchors each head $k$ to its fixed, random initialization $\widetilde{\phi}_k$. This prevents the ensemble from collapsing to a single solution, thereby preserving the diversity required for uncertainty estimation. As this is most relevant during early stages of training, where gradients tend to be large, but less relevant in later stages, the $\zeta$ parameter decays exponentially over the iterations of ACTIVEULTRAFEEDBACK. For a complete list of training hyperparameters, see Section F.3.

## D. Response Pair Selection Methods

This section explains the response pair selection algorithms from Section 4.3 in detail. For simplicity in notation, we drop the indexing by $i$ and consider a single prompt $x$ only. Let $\{y_j\}_{j=1}^m$ be the responses to $x$, and denote the corresponding lower and upper bounds of the reward estimate as vectors by $\underline{r}$ and $\overline{r}$.

**INFOMAX** (Saha, 2021) focuses purely on exploration with a goal to reduce uncertainty uniformly; therefore, it selects the ordered pair $(j, j')$ with $j \neq j'$ that maximizes the width of the confidence interval on the preference probability, $\arg\max_{j \neq j'} \overline{p}(y_j \succ y_{j'}) - \underline{p}(y_j \succ y_{j'})$, ignoring predicted reward quality.

---

**Algorithm 1** INFOMAX

---

1: **function** InfoMax($\underline{p}, \overline{p}$)
2:     Compute $\Delta_{j,j'}$ for all $j, j' \in \{1, \ldots, m\}$
3:     $\Delta_{j,j'} \leftarrow \begin{cases} -\infty, & j = j', \\ \overline{p}(y_j \succ y_{j'}) - \underline{p}(y_j \succ y_{j'}), & j \neq j'. \end{cases}$
4:     **return** $\arg\max_{(j,j')} \Delta_{j,j'}$
5: **end function**

---

**DOUBLE THOMPSON SAMPLING (DTS)** (Wu & Liu, 2016) balances exploration-exploitation by sampling a perturbed utility score for each response uniformly between its lower bound $\underline{r}$ and upper bound $\overline{r}$ and choosing the top response $y_j$; the second response $y_{j'}$ is obtained by resampling until $j' \neq j$ (up to `maxiter`) with a uniform-random fallback.

---

**Algorithm 2** DOUBLE THOMPSON SAMPLING (DTS)

---

1: **function** DTS($\underline{r}, \overline{r}$, maxiter)
2:     $j \leftarrow$ THOMPSONSAMPLE($\underline{r}, \overline{r}$) {first draw}
3:     **for** $t = 1$ to `maxiter` **do**
4:       $j' \leftarrow$ THOMPSONSAMPLE($\underline{r}, \overline{r}$); {resample until distinct}
5:       **if** $j \neq j'$ **then**
6:         **return** $(j, j')$
7:       **end if**
8:     **end for**
9:     **return** $(j, \text{Unif}(\{1, \ldots, m\} \setminus \{j\}))$ {fallback after `maxiter` resamples}
10: **end function**

---

**MAXMINLCB** (Pásztor et al., 2024) is based on pairwise lower confidence bounds (Equation (4)). It selects $j_1 = \arg\max_j \min_{j' \neq j} \underline{p}(y_j \succ y_{j'})$ to maximize the worst-case LCB against any opponent, and then $j_2 = \arg\min_{j \neq j_1} \underline{p}(y_{j_1} \succ y_j)$ to identify the opponent with the smallest LCB against $j_1$. We use $\epsilon$ for random tie-breaking among near-equal values (within $\epsilon$).

---

**Algorithm 3** MAXMINLCB

---

1: **function** MaxMinLCB$(\underline{p}, \overline{p}, \epsilon)$

2: $\quad L_{j,j'} \leftarrow \begin{cases} -\infty, & j = j' \\ \underline{p}(y_j \succ y_{j'}), & j \neq j' \end{cases} \quad \forall j, j' \in \{1, \ldots, m\}$ {ignore self/filtered pairs}

3: $\quad m_j \leftarrow \min_{j \neq j'} L_{j,j'} \; \forall j$ {worst-case LCB for each $j$}

4: $\quad j_1 \leftarrow$ RANDOMTIEBREAK$\{j : |m_j - \max_{j'} m_{j'}| < \epsilon\}$ {$\epsilon$-ties on maximin}

5: $\quad j_2 \leftarrow$ RANDOMTIEBREAK$\{j \neq j_1 : |L_{j_1,j} - \min_{j' \neq j_1} L_{j_1,j'}| < \epsilon\}$ {$\epsilon$-ties on argmin}

6: $\quad$ **return** $(j_1, j_2)$ {(chosen, rejected)}

7: **end function**

---

**DOUBLE REVERSED THOMPSON SAMPLING (DRTS)** extends DTS by drawing two independent Thompson samples, uniformly between the lower bound $\underline{r}$ and upper bound $\overline{r}$ for each response, and selecting the best and worst responses under these samples, respectively. This targets response pairs with a large expected quality gap while preserving the exploration benefits of Thompson sampling-based methods (e.g., occasionally selecting uncertain options). The parameter `maxiter` is the maximum number of resamples used to obtain $j' \neq j$ before falling back to a uniform draw over $\{1, \ldots, m\}$.

---

**Algorithm 4** DOUBLE REVERSED THOMPSON SAMPLING (DRTS)

---

1: **function** DRTS$(\underline{r}, \overline{r}, \text{maxiter})$

2: $\quad j \leftarrow$ THOMPSONSAMPLE$(\underline{r}, \overline{r})$ {sampled best}

3: $\quad$ **for** $t = 1$ to `maxiter` **do**

4: $\quad\quad j' \leftarrow$ THOMPSONSAMPLE$(-\overline{r}, -\underline{r})$ {sampled worst via reward reversal}

5: $\quad\quad$ **if** $j' \neq j$ **then**

6: $\quad\quad\quad$ **return** $(j, j')$

7: $\quad\quad$ **end if**{try to ensure a distinct pair}

8: $\quad$ **end for**

9: $\quad$ **return** $(j, \text{Unif}(\{1, \ldots, m\} \setminus \{j\}))$ {fallback after `maxiter` resamples}

10: **end function**

---

**DELTAUCB** selects an ordered response pair by maximizing the upper confidence bound on the preference probability. Thus, DELTAUCB deterministically targets the most optimistically likely win under the current confidence intervals. By relying on optimistic bounds rather than stochastic sampling, DELTAUCB steers exploration toward pairs that could plausibly exhibit substantial quality differences under uncertainty, while remaining fully deterministic given the current confidence intervals.

---

**Algorithm 5** DELTAUCB

---

1: **function** DeltaUCB$(\overline{p})$

2: $\quad \Delta_{j,j'} \leftarrow \begin{cases} -\infty, & j = j' \\ \overline{p}(y_{i,j} \succ y_{i,j'}), & j \neq j' \end{cases} \quad \forall j, j' \in \{1, \ldots, m\}$ {optimistic gap; forbid self-pairs}

3: $\quad$ **return** $\arg\max_{(j,j')} \Delta_{j,j'}$ {most optimistic win probability}

4: **end function**

---

## E. Annotation

Given the high cost and latency of human annotation at the scale required for our experiments, we opted to use an LLM-as-a-Judge approach. Specifically, we utilize Qwen 3 235B A22B[5] to score each response. In the following, we describe how we use the LLM to score each response (Section E.1) and ablate on the choice of Qwen 3 235B A22B, comparing it to models of different scales (Section E.2).

---

[5]Qwen/Qwen3-235B-A22B

### E.1. Scoring Methodology

Following recent findings (Ivison et al., 2024) that per-aspect annotation is most effective for synthetic data, we utilize the aspect-wise annotation proposed in UltraFeedback (Cui et al., 2024), using the aspects: $\mathcal{A} = \{\text{helpfulness, truthfulness, honesty, instruction following}\}$. Specifically, we prompt our LLM-as-a-Judge for each of these aspects, using varying system prompts to guide the model to score the response for this aspect. For the full prompt templates for each aspect, we refer the reader to Section H.2.

We explicitly instruct the LLM judge to output only the raw score as a single integer between 1 and 5, strictly suppressing any reasoning or chain-of-thought text. This strict output constraint allows us to calculate the aspect score $s_{\text{aspect}}$ by computing a softmax exclusively over the logits corresponding to the tokens for the digits 1 through 5. Given a prompt $x$, a response $y$, and the judging prompt $z_{x,y,\text{aspect}}$, the score is computed as:

$$s_{\text{aspect}}(y \mid x) = \sum_{k=1}^{5} k \cdot \frac{\exp\left(\ell_k(z_{x,y,\text{aspect}})\right)}{\sum_{j=1}^{5} \exp\left(\ell_j(z_{x,y,\text{aspect}})\right)}$$

where $\ell_k(z_{x,y,\text{aspect}})$ denotes the logit output by the judge for the token corresponding to integer $k$ when given the input prompt $z_{x,y,\text{aspect}}$.

The final scalar quality score for the response is then obtained by averaging over the set of aspects:

$$s_{\text{overall}}(y \mid x) = \frac{1}{|\mathcal{A}|} \sum_{\text{aspect}\in\mathcal{A}} s_{\text{aspect}}(y \mid x).$$

Crucially, this continuous scoring mechanism addresses the issue of score saturation. We attribute such saturation to the inherent numeric bias of LLMs, where models disproportionately favor higher integers (e.g., 5). This tendency renders competitive responses indistinguishable when using discrete labels. By utilizing the expected value over token probabilities, we capture the judge's underlying confidence, enabling fine-grained ranking even among responses with identical discrete scores.

*Table 4.* Comparison of the four experimental judging configurations using the Qwen/Qwen3-235B-A22B model on the **UltraFeedback** dataset ($N = 60'829$). Win Rate measures the percentage of samples where the judge assigned a strictly higher overall score to the preferred response. Ties occur when the calculated overall score is identical for both responses. The *Probabilistic Scoring* configuration (without reasoning) was selected for the final annotation pipeline due to its superior alignment, reliability, and speed.

| Configuration | Win Rate | Tie Rate | Parse Errors |
|---|---|---|---|
| **Probabilistic Scoring** | **76.70%** | **0.0%** | **0** |
| Discrete Generation | 75.36% | 14.7% | 275 |
| Probabilistic Scoring + Explicit Reasoning | 73.54% | 11.3% | 120 |
| Discrete Generation + Explicit Reasoning | 73.37% | 12.1% | 20,181 |

This necessity for a distributed signal drove the decision to suppress the model's explicit reasoning capabilities. As shown in Table 4, our experiments on the **UltraFeedback** prompts in combination with responses from our model pool (Section B.1) reveal that enabling reasoning degrades performance across both scoring methods. We observed that when the judge reasons, it becomes overly certain, collapsing the probability distribution over score tokens into a single peak (score saturation). In fact, the analysis confirms that with reasoning enabled, approximately 88.4% (53'763/60'829) of the prompts resulted in a strict probability of 1.0 assigned to a single integer token for every aspect of both responses[6]. This effectively reverts the continuous signal to a discrete integer, lowering the win rate to 73.54%. In contrast, the *Probabilistic Scoring* configuration consistently maintained a distributed probability mass, avoiding collapse entirely. This preservation of uncertainty allowed this method to distinguish between competitive responses, eliminating ties and achieving a superior win rate of 76.70%, effectively outperforming the 75.36% achieved by the discrete generation variant.

---

[6]We utilized vLLM (Kwon et al., 2023) for inference, configured to return the top-20 log probabilities. In these instances, only one of the target integer tokens (1–5) appeared within the top-20 candidates. This implies that the logits for the remaining score tokens were negligible, resulting in a renormalized probability of 1.0 for the top token.

The *Probabilistic Scoring* strategy additionally encourages validity. While the *Discrete Generation + Explicit Reasoning* setup suffered over 20'000 parsing failures (out of ~486'000 total inference calls) due to format deviations, the selected probabilistic approach yielded zero errors across all samples. Additionally, suppressing the reasoning step resulted in a massive gain in inference throughput, operating at approximately $15\times$ the speed of the reasoning-enabled configurations (~12'000 vs. ~800 samples/hr).

The parsing errors observed when scoring with Discrete Generation could have been avoided by taking the score value with highest logit. This configuration, which we call Discrete Scoring, is slightly simpler than Probabilistic Scoring, which involves taking an expectation over the score distribution. Even so, we ultimately opted for Probabilistic Scoring because we found it to perform better on RewardBench 2 (Malik et al., 2025) – a benchmark that has high correlation with downstream performance – in particular on the Ties subtask for judge sensitivity to similar answers (see Table 5). Further comparisons on the AlpacaEval 2.0 benchmark (Dubois et al., 2024) also revealed that while both scoring methodologies led to similarly high correlations with human annotations, our Probabilistic Scoring methodology achieves higher human agreement (see Table 6). Altogether, these results support our LLM-as-a-Judge setup for the main experiments.

*Table 5.* Comparison of Probabilistic vs Discrete Scoring using Qwen/Qwen3-235B-A22B on RewardBench 2 (Malik et al., 2025).

| Configuration | Tie Rate | Mean |
|---|---|---|
| **Probabilistic Scoring** | **0.833** | **0.744** |
| Discrete Scoring | 0.729 | 0.698 |

*Table 6.* Comparison of Probabilistic vs Discrete Scoring using Qwen/Qwen3-235B-A22B on AlpacaEval 2.0 (Dubois et al., 2024).

| Configuration | Spearman's $r$ | Human Agreement |
|---|---|---|
| **Probabilistic Scoring** | 0.67 | **62.7%** |
| Discrete Scoring | 0.70 | 60.4% |

### E.2. Judge Model Ablation

To evaluate the effectiveness of our LLM-as-a-Judge design, we evaluate our choice of judge model on RewardBench 2 (Malik et al., 2025). The results can be seen in Table 7.

Our judge approach performs similarly for all models, yielding accurate scores. It is important to note that while Skywork-Reward-V2-Llama-3.1-8B achieves a superior score on RewardBench 2, using its rewards as annotation scores resulted in significant degradation of the fine-tuned models in our early experiments, motivating us to opt for our judge instead. Because of this, we opted to use Qwen 3 235B A22B throughout our experiments, for its strong performance for reward modeling and general fine-tuning.

## F. Implementation Details

### F.1. Evaluation Methodology

To assess the quality of the datasets generated by ACTIVEULTRAFEEDBACK, we conduct experiments targeting both stages of the standard RLHF pipeline (Section 3): reward modeling and policy optimization. By evaluating these components in isolation, we can disentangle the data's impact on both stages. It is important to note that the models trained for evaluation are distinct from the ENN reward model utilized within the ACTIVEULTRAFEEDBACK acquisition loop.

For both reward modeling and fine-tuning experiments, we utilize Llama-3.1-Tulu-3-8B-SFT[7] (Lambert et al., 2025) as the base model and use parameter-efficient fine-tuning via LoRA adapters (Hu et al., 2022) and the AdamW optimizer (Loshchilov & Hutter, 2019) for all training runs.

The objectives for both trainings follow standard procedures, using the Bradley-Terry objective (Equation (1)) for reward modeling and direct preference optimization (DPO) (Rafailov et al., 2023) for fine-tuning.

---

[7]allenai/Llama-3.1-Tulu-3-8B-SFT

## F.2. Training Stability

In this section, we analyze the stability of ACTIVEULTRAFEEDBACK and our evaluation setup. In order to analyse the stability of ACTIVEULTRAFEEDBACK, we keep the responses and annotation scores fixed, to conserve computational resources (Section F.4), and evaluate the stability of the response pair acquisition and ENN training in ACTIVEULTRAFEEDBACK. For this, we consider two response pair selection methods. One deterministic method (DELTAUCB) and one sampling-based method (DRTS) to also evaluate the stability of sampling-based methods. The results can be seen in Table 8.

*Table 8.* Stability of ACTIVEULTRAFEEDBACK across 5 different random seeds with two response pair selection methods. We report the mean and standard deviation for each benchmark. Scores are reported as relative deltas to the base model.

| Method | GSM8K | IFEval | TruthfulQA | AlpacaEval 2 | Mean | RewardBench 2 |
|---|---|---|---|---|---|---|
| DRTS | $+0.057_{\pm0.009}$ | $+0.025_{\pm0.017}$ | $+0.132_{\pm0.010}$ | $+0.246_{\pm0.007}$ | $+0.114_{\pm0.006}$ | $+0.277_{\pm0.025}$ |
| DELTAUCB | $+0.058_{\pm0.009}$ | $+0.017_{\pm0.009}$ | $+0.103_{\pm0.007}$ | $+0.230_{\pm0.012}$ | $+0.101_{\pm0.006}$ | $+0.282_{\pm0.011}$ |

We observe that, for downstream evaluations, both deterministic and sampling-based methods are very stable, only having a standard deviation of 0.006 in their mean downstream score. For reward modelling, the sampling-based methods experience slightly higher standard deviation (0.025) than the deterministic methods (0.011), which is to be expected when introducing more stochasticity through sampling.

Now we analyse the stability of our evaluation setup, starting with the DPO training. We utilize the decontaminated version of the **UltraFeedback** dataset[8] (Cui et al., 2024) for these experiments. First, we examine the sensitivity to initialization by training with 5 different random seeds while keeping all other hyperparameters fixed. We ensure reproducibility by fixing the random seed and explicitly shuffling the dataset according to the seed before training.

As shown in Table 9, the standard deviation across seeds is minimal ($\approx 0.003$ for the overall score), with TruthfulQA exhibiting the highest stability (0.001) and AlpacaEval 2 showing slightly higher variance (0.006), likely due to the inherent noise in generation-based evaluation.

*Table 9.* Training stability across 5 different random seeds. We report the mean and standard deviation for each benchmark. Scores are reported as relative deltas to the base model.

| Metric | GSM8K | IFEval | TruthfulQA | AlpacaEval 2 | Mean |
|---|---|---|---|---|---|
| Mean | +0.039 | +0.020 | +0.056 | +0.028 | +0.035 |
| Std. Dev. | 0.005 | 0.006 | 0.001 | 0.006 | 0.003 |

Next, to assess the inherent randomness caused by system-level non-determinism (e.g., PyTorch non-determinism, and non-associativity of rounding operations for floating-point numbers in multi-GPU setups), we performed 5 independent training runs using a fixed seed of 42. The results in Table 10 confirm that system-level noise produces deviations comparable to seed variation ($\approx 0.004$ overall). IFEval shows slightly higher variance here (0.011), while TruthfulQA remains perfectly stable.

*Table 10.* Training stability across 5 runs with a fixed seed (Seed 42), assessing system-level non-determinism. Scores are reported as relative deltas to the base model.

| Metric | GSM8K | IFEval | TruthfulQA | AlpacaEval 2 | Mean |
|---|---|---|---|---|---|
| Mean | +0.044 | +0.020 | +0.054 | +0.030 | +0.035 |
| Std. Dev. | 0.003 | 0.011 | 0.000 | 0.008 | 0.004 |

We performed the same stability analysis for our Reward Model training using RewardBench 2. First, examining initialization sensitivity across 5 random seeds (Table 11), we observe moderate stability overall ($\approx 0.011$). However, the Ties metric

---

[8]allenai/ultrafeedback_binarized_cleaned

exhibits significant variance (0.072), indicating that the model's ability to resolve subtle preference differences is highly sensitive to random initialization conditions.

*Table 11.* Reward Model training stability across 5 different random seeds. Scores are reported as relative deltas to the base model.

| Metric | Factuality | Focus | Math | Precise IF | Safety | Ties | Mean |
|---|---|---|---|---|---|---|---|
| Mean | +0.344 | +0.495 | +0.145 | +0.095 | +0.453 | +0.253 | +0.298 |
| Std. Dev. | 0.019 | 0.029 | 0.030 | 0.031 | 0.036 | 0.072 | 0.011 |

Second, we performed 5 independent training runs using a fixed seed of 42. The results in Table 12 reveal negligible noise ($\approx 0.004$). Notably, the Ties variance drops to 0.008, confirming that the higher instability observed previously stems from algorithmic randomness (e.g., weight initialization, data permutation) rather than hardware-level non-determinism.

*Table 12.* Reward Model stability across 5 runs with a fixed seed (Seed 42). Scores are reported as relative deltas to the base model.

| Metric | Factuality | Focus | Math | Precise IF | Safety | Ties | Mean |
|---|---|---|---|---|---|---|---|
| Mean | +0.363 | +0.444 | +0.145 | +0.128 | +0.546 | +0.252 | +0.292 |
| Std. Dev. | 0.005 | 0.006 | 0.007 | 0.007 | 0.006 | 0.008 | 0.004 |

Finally, we extend our stability analysis to the optimization algorithms themselves. To ensure that our performance gains are robust and not artifacts of initialization, we trained both IPO and SimPO models using 5 different random seeds. As detailed in Tables 13 and 14, our setup proves to be highly stable across different preference optimization algorithms. Both methods demonstrate minimal variance across key benchmarks (e.g., standard deviations of $\approx 0.004$–$0.011$ on GSM8K and $\approx 0.005$–$0.006$ on TruthfulQA). These results, reflected in the low variance of the aggregated mean scores (0.015 for SimPO and 0.011 for IPO), confirm that the improvements over the baseline are reliable and consistent.

*Table 13.* Stability analysis of our SimPO algorithms setup. We report the Mean and Standard Deviation across 5 different random seeds. Scores are reported as relative deltas to the base model.

| Benchmark | GSM8K | IFEval | TruthfulQA | AlpacaEval | Mean |
|---|---|---|---|---|---|
| Mean Delta | +0.033 | +0.019 | +0.058 | +0.273 | +0.095 |
| Std. Dev. | 0.011 | 0.009 | 0.006 | 0.036 | 0.015 |

*Table 14.* Stability analysis of our IPO algorithms setup. We report the Mean and Standard Deviation across 5 different random seeds. Scores are reported as relative deltas to the base model.

| Benchmark | GSM8K | IFEval | TruthfulQA | AlpacaEval | Mean |
|---|---|---|---|---|---|
| Mean Delta | +0.048 | +0.035 | +0.040 | +0.304 | +0.106 |
| Std. Dev. | 0.004 | 0.005 | 0.005 | 0.036 | 0.011 |

### F.3. Hyperparameters

Throughout our work, we have conducted extensive experiments for identifying well-performing and robust hyperparameters for different modules of our pipeline, including: response generation, annotation pipeline, ENN reward model, several direct preference optimization algorithms, and reward model training. In this section, we detail all hyperparameters along with their final values, and, if applicable, the sweep range we used to identify the final values.

**Batch Size**  The number of prompts per iteration of ACTIVEULTRAFEEDBACK is fixed at 64 for all experiments.

**Response Generation and Annotation**  We use vLLM (Kwon et al., 2023) for prompting LLMs in two stages of the ACTIVEULTRAFEEDBACK pipeline: Response Generation (Section 4.1) and Preference Annotation (Section 4.4). The sampling parameters used for each stage are listed in Table 15.

**ENN Reward Model**  The hyperparameters for the ENN reward model in the Reward Prediction stage of ACTIVEULTRAFEEDBACK (Section 4.2) are listed in Table 16. Most values are adopted from prior work (Dwaracherla et al., 2024). As a base model for the ENN reward model, we use Skywork Reward V2 Qwen3 4B[9] for its strong reward modelling performance, and train the MLP head ensemble on the last-layer embedding of the last token in the sequence.

**ENN Training**  The Reward Model Training component of ACTIVEULTRAFEEDBACK (Section 4.5) involves many hyperparameters. We list the ones that are fixed across all experiments in Table 17.

*Table 17.* Fixed hyperparameters used across experiments for ENN training.

| Hyperparameter | Value |
|---|---|
| Max Length (Prompt + Response) | 4096 |
| Batch Size, $\mathcal{B}$ | 64 |
| Train Steps | 100 |
| Initial Regularization, $\zeta$ | 1.0 |
| Reward Centering Coefficient, $\gamma$ | 0.01 |
| Learning Rate | $5 \times 10^{-5}$ |

For certain hyperparameters, the optimal value differs based on the active response pair selection method, as well as between DPO fine-tuning and reward modeling. We report the sweep performed and the optimal configuration we found in Table 18. The sweep budget was identical across all ENN-based active selection methods. The static baselines (RANDOM, ULTRAFEEDBACK, MAXMIN) do not utilize the ENN and are therefore excluded from this ENN-specific sweep.

**Preference Optimization (DPO, IPO, SimPO)**  To establish the optimal configuration for preference fine-tuning, we utilized the UltraFeedback dataset[10] (Cui et al., 2024). We conducted a hyperparameter sweep for DPO, IPO, and SimPO and selected based on best performance in our evaluation framework (Section F.1), are presented in Table 19. We fixed the batch size to 32, used a linear learning rate schedule with a warmup ratio of 0.1, and used a max length (prompt + completion) of 2048 for all three preference optimization algorithms.

**Reward Modeling**  The hyperparameter sweep and final values for reward model training, selected based on the highest mean score on RewardBench 2, are listed in Table 20. We fixed the BATCH SIZE to 128, used a constant learning rate, and used a max length (prompt + completion) of 4096.

**LoRA**  We use the hyperparameters in Table 21 for LoRA when fine-tuning (DPO, IPO, SimPO) and reward modeling.

### F.4. Compute Estimates

All experiments were conducted on 8 NVIDIA GH200 Grace Hopper Superchips. To facilitate extensive ablation studies and rapid iteration, we decoupled the computationally expensive generation and annotation phases from the active learning loop. Specifically, we pre-computed the candidate responses and their corresponding judge annotations for the entire dataset prior to simulating the acquisition process.

Table 22 provides a breakdown of the estimated GPU hours required for each stage of the pipeline on the **UltraFeedback** dataset. As shown, the computational budget is roughly evenly distributed between response generation and the pre-computation of judge scores. In practical use of ACTIVEULTRAFEEDBACK, the annotation cost would be drastically reduced, as the pipeline only requires annotations for the selected responses, rather than the entire candidate pool.

---

[9]Skywork/Skywork-Reward-V2-Qwen3-4B
[10]allenai/ultrafeedback_binarized_cleaned

It is important to note that our implementation prioritized experimental flexibility and reproducibility over maximum computational efficiency. Compared to static pipelines such as ULTRAFEEDBACK, our current implementation trades higher upfront response-generation compute for improved sample efficiency and lower annotation demand, since we pre-computed candidate responses and judge annotations for the full dataset before simulating acquisition. In practical deployment, a strong selection strategy would require processing substantially fewer selected prompts end-to-end. Moreover, Section G.6 shows that competitive performance can be retained with smaller yet more diverse response model pools, suggesting a concrete path to reducing response generation cost. Further runtime reductions are also likely through more optimized distributed inference and training configurations. In total, all experiments, including model fine-tuning, reward model training, ablations, stability analyses, failed experiments, and preliminary experiments, consumed approximately 200'000 GPU hours.

## G. Additional Results

### G.1. Generated Dataset Analysis

To understand the selection dynamics of different response pair acquisition methods, we analyze the distributions of the generated datasets by examining how often each model from our pool was selected, how often it was annotated as chosen and rejected (Figure 6) and the mean scores for the chosen and rejected responses for different response pair selection methods (Table 23).

We find that methods aiming at regret minimization, such as DTS (Figure 6e) and MAXMINLCB (Figure 6f), successfully identify high-quality models, with high judge scores (Table 23), resulting in distributions heavily skewed towards recent, large-scale models. This highlights the key distinction between identifying strong responses and creating informative preference pairs: for downstream preference learning, high-quality chosen responses are not sufficient on their own if the rejected responses are also too strong, since this reduces the effective supervision signal. In contrast, as expected, RANDOM (Figure 6a) exhibits a nearly uniform distribution, while ULTRAFEEDBACK (Figure 6b) displays a slight skew towards higher-quality models due to its "best-of-$N$" heuristic. Conversely, the entropy-minimizing INFOMAX (Figure 6d) disproportionately selects smaller, older models. We attribute this to the fact that recent, large-scale models consistently achieve near-perfect scores, leading to high certainty in their high quality. In contrast, smaller models exhibit erratic behaviour, occasionally producing high-scoring responses but frequently failing. This unpredictability results in higher epistemic uncertainty, driving the method to sample from them more frequently. Finally, our proposed quality delta maximization methods, DRTS (Figure 6g) and DELTAUCB, produce distributions closely mirroring the high-scoring, but inefficient MAXMIN baseline (Figure 6c), prioritizing the best and worst responses, yet achieve this efficiently by requiring only two annotations per prompt compared to MAXMIN's annotation of the full candidate set.

*Table 23.* Mean score of the chosen, rejected, and overall responses from different response pair selection methods on the **UltraFeedback** prompts.

| Method | Mean Chosen Score | Mean Rejected Score | Mean Score |
|---|---|---|---|
| RANDOM | 4.522 | 3.564 | 4.043 |
| ULTRAFEEDBACK | 4.747 | 3.810 | 4.279 |
| MAXMIN | 4.925 | 1.605 | 3.625 |
| DELTAQWEN | 4.549 | 2.924 | 3.736 |
| INFOMAX | 3.666 | 3.156 | 3.411 |
| DTS | 4.855 | 4.584 | 4.720 |
| MAXMINLCB | 4.864 | 4.683 | 4.773 |
| DRTS | 4.752 | 1.968 | 3.360 |
| DELTAUCB | 4.705 | 2.113 | 3.409 |

### G.2. Sample Efficiency without AlpacaEval 2

The score deltas in AlpacaEval 2 are an order of magnitude larger than those in our other benchmarks. Consequently, the mean score delta is disproportionately influenced by AlpacaEval 2, obscuring performance trends in the wider suite. To

provide a clearer visualization of our sample efficiency experiment (Section 5.3), Figure 7 presents the mean performance trajectories both with and without the inclusion of AlpacaEval 2.

### G.3. Full Input Prompt Dataset Ablation

In this section, we provide the detailed scores for our prompt dataset ablation (Section 5.4). The detailed results, for each individual benchmark and response pair selection method, can be seen in Table 24.

### G.4. Full Preference Optimization Algorithm Ablation

In this section, we provide the detailed scores for our preference optimization algorithm ablation (Section 5.5). The detailed results, for each individual benchmark and response pair selection method, can be seen in Table 25.

### G.5. Full Response Pair Selection Method Sample Efficiency Ablation

In addition to the partial visualization of the sample efficiency of different response pair selection methods in Figure 3 and Figure 5, we provide a full visualization comparing all methods for DPO and reward modelling in Figure 8 and for IPO and SimPO in Figure 9.

### G.6. Response Pool Size and Diversity Ablation

To assess the sensitivity of ACTIVEULTRAFEEDBACK to the size and diversity of the response model pool, we run DRTS on four modified pools grouped by parameter scale: the full 30-model pool, the 15 largest models only, the 15 smallest models only, and a diverse 10-model pool containing the 5 largest and 5 smallest models. We then fine-tune with DPO on each resulting dataset and evaluate downstream performance.

The results in Table 26 show that diversity matters more than pool size alone. Restricting the pool to the 15 largest models substantially hurts performance, suggesting that large models by themselves do not provide sufficiently informative variation. In contrast, the 15 smallest-model pool remains reasonably effective, likely because it still spans meaningful quality differences. Importantly, the smaller but diverse pool of 5 largest and 5 smallest models remains competitive with the full 30-model pool, indicating that strong performance can be retained with fewer generations when the pool preserves diversity.

## H. Prompt Templates

In this section, we provide the prompt templates used in our pipeline for both the response generation (Section 4.1) and preference annotation (Section 4.4). All of the prompts used have been originally taken from UltraFeedback (Cui et al., 2024).

### H.1. Response Generation Prompt Templates

For each response, we randomly sample a principle among "helpfulness", "truthfulness", and "honesty". For each of these principles we use 11 different system prompts and provide one representative system prompt here. You can find all prompts in our open-sourced code.

---

**Response Generation Prompt Template Examples**

**Helpfulness** : The assistant should provide users with accurate, relevant, and up-to-date information, ensuring that the content is positive, interesting, engaging, educational, and helpful.

**Truthfulness** : The assistant should be honest about whether it knows the answer and express its uncertainty explicitly. Be confident on questions it knows well and be modest on those it is unfamiliar with. Use weakeners such as 'I guess', 'I suppose', 'probably', and 'perhaps' to express uncertainty, and feel free to answer 'I don't know' if necessary.

**Honesty** : The assistant should answer truthfully and be faithful to factual knowledge as well as given contexts, never making up any new facts that aren't true or cannot be grounded in the instruction.

---

### H.2. Annotation Prompt Templates

Our annotation setup utilizes a single shared system prompt for all annotations to enforce the role of an impartial judge and strict output formatting. The following system prompt is used for all aspects to ensure the judge outputs only a single integer score.

---

**Judge System Prompt**

You are an impartial judge. Your role is to critically evaluate the quality of an AI assistant response based on a given criteria. You'll receive an input with two sections, enclosed in tags: `<INSTRUCTIONS>...</INSTRUCTIONS>` for the instructions, and `<TEXT>...</TEXT>` for the AI assistant response. Carefully read and interpret the instructions to assess how well the text fulfills the criteria requirements. You will be given a scoring rubric below, based on which you should provide a rating from 1 to 5. Your output should only be an integer from 1 to 5. Do not output any additional text or explanations.

---

For the user prompt, we construct a specific rubric based on the aspect being evaluated ("instruction following", "honesty", "truthfulness", or "helpfulness"). The final user prompt is constructed by using these rubrics and injecting the original prompt (`{prompt}`) and the response to be evaluated by the LLM judge (`{response}`).

---

**Instruction Following Annotation Prompt Template**

You will be doing an Instruction Following Assessment of an AI assistant response.
Carefully read and interpret instructions to assess how well the text fulfills the task requirements.
**Scoring**: Rate the text on a scale of 1 to 5:

1. **Irrelevant**: No alignment.

2. **Partial Focus**: Addresses one aspect poorly.

3. **Partial Compliance**:

   - (1) Meets goal or restrictions, neglecting other.
   - (2) Acknowledges both but slight deviations.

4. **Almost There**: Near alignment, minor deviations.

5. **Comprehensive Compliance**: Fully aligns, meets all requirements.

Here are the instructions and the AI assistant response:
`<INSTRUCTIONS>{prompt}</INSTRUCTIONS>`
`<TEXT>{response}</TEXT>`

---

---

**Honesty Annotation Prompt Template**

You will be doing a Honesty and Uncertainty Expression Assessment of an AI assistant response.
Assess how well the text conveys honesty and uncertainty. Evaluate if the text's confidence aligns with response correctness.

- **Indicators of Uncertainty**:
    1. Weakeners: e.g., 'I guess,' 'probably.'
    2. Refusal to answer.
    3. Verbalized confidence scores: [0, 20] low; (20, 40] uncertain; (40, 60] moderate; (60, 80] leaning confident; (80, 100] high.

- **No uncertainty expression indicate confidence.**

- **Response Correctness**: Align with ground truth, or provide accurate content without fabrication.

**Scoring**: Rate outputs on a scale of 1 to 5:

1. **Confidently Incorrect**: Confident but entirely wrong.

2. **Confident with Significant Mistakes / Unconfident Incorrect**:

    - Confident but contains major errors.
    - Unconfident and entirely wrong.

3. **Uncertain / 'I Don't Know' / Subtle Mistakes**:

    - 'I don't know' or declines.
    - Confident but contains minor errors.
    - Unconfident and contains significant mistakes.

4. **Correct but Uncertain / Expressed Subtle Mistakes**:

    - Correct but unconfident.
    - Makes subtle mistakes but expresses uncertainty without specifying the exact area of doubt.

5. **Correct and Confident / Precisely Express Uncertainty**:

    - Correct and confident.
    - Makes mistakes, but precisely acknowledges minor errors and indicates uncertainty on potential mistakes.

Here are the instructions and the AI assistant response:
```
<INSTRUCTIONS>{prompt}</INSTRUCTIONS>
<TEXT>{response}</TEXT>
```

---

**Truthfulness Annotation Prompt Template**

You will be doing a Truthfulness and Hallucination Assessment of an AI assistant response.
Evaluate the text's accuracy in providing information without introducing misleading or fabricated details.
When evaluating truthfulness, consider the following types of hallucination:

1. **Contradictory with the World (Factual Error)**: Entities, locations, concepts, or events that conflict with established knowledge.

2. **Contradictory with Instruction and Input**: Responses diverge, introducing new facts not aligned with instructions or inputs.

3. **Self-Contradictory / Logical Error**: Responses contain internal contradictions or logical errors within each independent text.

Reflect on whether any of these hallucination types are present in the response, and take them into account when assigning your rating.
**Scoring**: Rate outputs on a scale of 1 to 5 based on extent of hallucination:

1. **Completely Hallucinated**: Entirely unreliable due to hallucinations.

2. **Severe Hallucination**: Nearly half contains hallucinations, severe deviation from main points.

3. **Partial Hallucination / Misunderstanding**: Overall truthful, partial misunderstanding due to hallucinations.

4. **Insignificant Hallucination**: Mostly truthful, slight hallucination not affecting main points.

5. **No Hallucination**: Free of hallucinations.

Here are the instructions and the AI assistant response:
<INSTRUCTIONS>{prompt}</INSTRUCTIONS>
<TEXT>{response}</TEXT>

---

---

**Helpfulness Annotation Prompt Template**

You will be doing an Informativeness / Helpfulness Assessment of an AI assistant response.
Evaluate if the text fulfills task objectives and provides high-quality, correct, and, informative content.
Helpfulness assessment emphasizes **Overall Quality** regarding correctness and informativeness.
**Correctness**: Accurate computation, reasoning steps, and outputs without misunderstandings or fabrication.
When assessing informativeness, consider the following aspects:

1. **Clarity and Relevance**: Does the response relate to the task and seek clarifications if needed?

2. **Useful and Comprehensive Information**: Does it provide relevant background, reasoning steps, or detailed description?

3. **Not Lengthy, No Repetition**: Is the response concise, avoiding verbosity or repetition?

Score on a scale of 1 to 5 based on extent of helpfulness, regarding both informativeness and correctness:

1. **Severely Incorrect**: Contains significant inaccuracies or fabricated content, even if comprehensive information is provided.

2. **Partially Incorrect**: Contains errors that may cause confusion, even though comprehensive information is present.

3. **Correct**: Accurate and provides useful information that meets the task's requirements.

4. **Highly Informative**: Accurate and extensive, providing valuable insights and detailed information.

5. **Outstandingly Helpful**: Both accurate and in-depth, offering profound insights and comprehensive information.

Here are the instructions and the AI assistant response:
```
<INSTRUCTIONS>{prompt}</INSTRUCTIONS>
<TEXT>{response}</TEXT>
```

---

*Table 3.* The 30 models used for response generation with their total number of parameters (in billions) and licenses. Separators are placed between models from different families.

| Model | # Parameters | License |
|---|---|---|
| Qwen/Qwen2.5-0.5B-Instruct | 0.5B | Apache 2.0 |
| Qwen/Qwen2.5-72B-Instruct | 72B | Qwen |
| Qwen/Qwen3-0.6B | 0.6B | Apache 2.0 |
| Qwen/Qwen3-1.7B | 1.7B | Apache 2.0 |
| Qwen/Qwen3-14B | 14B | Apache 2.0 |
| Qwen/Qwen3-30B-A3B | 30B | Apache 2.0 |
| Qwen/Qwen3-32B | 32B | Apache 2.0 |
| Qwen/Qwen3-235B-A22B | 235B | Apache 2.0 |
| meta-llama/Llama-3.1-8B-Instruct | 8B | Llama 3 |
| meta-llama/Llama-3.2-1B-Instruct | 1B | Llama 3 |
| meta-llama/Llama-3.2-3B-Instruct | 3B | Llama 3 |
| meta-llama/Llama-3.3-70B-Instruct | 70B | Llama 3 |
| microsoft/Phi-4-mini-instruct | 4B | MIT |
| microsoft/phi-4 | 14B | MIT |
| mistralai/Mistral-Small-24B-Instruct-2501 | 23B | Apache 2.0 |
| mistralai/Mistral-Large-Instruct-2411 | 123B | MRL |
| nvidia/Llama-3.1-Nemotron-70B-Instruct-HF | 70B | Llama 3 |
| nvidia/Llama-3_3-Nemotron-Super-49B-v1 | 49B | Nvidia Open Model |
| nvidia/Llama-3_1-Nemotron-Ultra-253B-v1 | 253B | Nvidia Open Model |
| google/gemma-3-1b-it | 1B | Gemma |
| google/gemma-3-4b-it | 4B | Gemma |
| google/gemma-3-12b-it | 12B | Gemma |
| google/gemma-3-27b-it | 27B | Gemma |
| allenai/OLMo-2-0325-32B-Instruct | 32B | Apache 2.0 |
| allenai/Llama-3.1-Tulu-3-70B | 70B | Llama 3 |
| allenai/Llama-3.1-Tulu-3-405B | 405B | Llama 3 |
| HuggingFaceT/SmolLM2-1.7B-Instruct | 1.7B | Apache 2.0 |
| moonshotai/Moonlight-16B-A3B-Instruct | 16B | MIT |
| CohereLabs/c4ai-command-a-03-2025 | 111B | CC by NC 4.0 |
| deepseek-ai/DeepSeek-V3 | 671B | Deepseek |

*Table 7.* Rewardbench 2 scores for our judge using different models as judge models. With this comparison, we aim to cover a wide range of model sizes to examine how model size affects annotation quality. We also added Skywork-Reward-V2-Llama-3.1-8B, the current rank 1 on RewardBench 2, as a reference.

| Model | Factuality | Focus | Math | Precise IF | Safety | Ties | Mean |
|---|---|---|---|---|---|---|---|
| Qwen3-32B | 0.787 | 0.840 | 0.710 | 0.343 | 0.844 | 0.863 | 0.731 |
| Qwen3-235B-A22B | 0.851 | 0.792 | 0.689 | 0.369 | 0.931 | 0.833 | 0.744 |
| Llama-3.3-70B-Instruct | 0.692 | 0.753 | 0.683 | 0.437 | 0.806 | 0.866 | 0.706 |
| Skywork-Reward-V2-Llama-3.1-8B | 0.844 | 0.983 | 0.770 | 0.656 | 0.967 | 0.812 | 0.839 |

*Table 15.* Sampling parameters for Response Generation and Preference Annotation in ACTIVEULTRAFEEDBACK.

| Hyperparameter | Response Generation | Preference Annotation |
|---|---|---|
| Temperature | 1.0 | 0.0 |
| Top-$p$ | 1.0 | – |
| Max Response Tokens | 4096 | 16 |

*Table 16.* Hyperparameters for the ENN architecture.

| Hyperparameter | Value |
|---|---|
| Number of MLP heads | 20 |
| Number of layers per MLP head | 2 |
| Hidden size of each MLP head | 128 |

*Table 18.* ENN training hyperparameters and sweep ranges for each active response pair selection method. Separate optimal values were chosen based on performance after DPO fine-tuning and on RewardBench 2.

| Hyperparameter | Grid Values | INFOMAX | DTS | MAXMINLCB | DRTS | DELTAUCB |
|---|---|---|---|---|---|---|
| **Optimal for DPO Fine-Tuning** | | | | | | |
| Beta $\beta$ | [1, 2] | 2 | 1 | 1 | 1 | 2 |
| Regularization Decay | [0.9, 0.99, 0.999] | 0.99 | 0.99 | 0.99 | 0.999 | 0.999 |
| Replay Buffer Size Factor, $\rho$ | [100, 1000] | 1000 | 1000 | 1000 | 1000 | 1000 |
| **Optimal for Reward Modeling** | | | | | | |
| Beta $\beta$ | [1, 2] | 2 | 1 | 2 | 1 | 1 |
| Regularization Decay | [0.9, 0.99, 0.999] | 0.99 | 0.999 | 0.9 | 0.9 | 0.99 |
| Replay Buffer Size Factor, $\rho$ | [100, 1000] | 1000 | 1000 | 1000 | 1000 | 100 |

*Table 19.* Optimal hyperparameters for our DPO, IPO, and SimPO fine-tuning, selected based on evaluation performance.

| Hyperparameter | Grid Values | Chosen Value |
|---|---|---|
| **For DPO** | | |
| Learning Rate | $[1 \times 10^{-6}, 2 \times 10^{-5}, 5 \times 10^{-4}]$ | $2 \times 10^{-5}$ |
| Lambda $\lambda$ | [0.1, 0.01] | 0.1 |
| Epochs | [1, 3] | 3 |
| **For IPO** | | |
| Learning Rate | $[5 \times 10^{-6}, 1 \times 10^{-5}, 2 \times 10^{-5}, 5 \times 10^{-5}]$ | $5 \times 10^{-6}$ |
| Lambda $\lambda$ | [0.01, 0.1, 0.5, 1.0] | 0.01 |
| Epochs | [1, 3] | 1 |
| **For SimPO** | | |
| Learning Rate | $[5 \times 10^{-6}, 1 \times 10^{-5}, 2 \times 10^{-5}, 5 \times 10^{-5}]$ | $5 \times 10^{-6}$ |
| Gamma | [0.3, 0.5, 1.0, 1.2, 1.4, 1.6] | 1.2 |
| Lambda $\lambda$ | [2.0, 2.5] | 2.0 |
| Epochs | [1, 3] | 1 |

*Table 20.* Optimal hyperparameters for reward model training, selected based on RewardBench 2 performance.

| Hyperparameter | Grid Values | Chosen Value |
|---|---|---|
| Learning Rate | $[3 \times 10^{-6}, 5 \times 10^{-6}, 2 \times 10^{-5}]$ | $2 \times 10^{-5}$ |
| Epochs | [1, 2, 3] | 2 |

*Table 21.* Hyperparameters for our LoRA setup.

| Hyperparameters | Chosen Value |
|---|---|
| Rank | 64 |
| Alpha | 16 |
| Dropout | 0.1 |
| Target Modules | all-linear |

| Step | Estimated GPU Hours |
|---|---|
| Response Generation | 600 |
| Annotation | 600 |
| Active Learning Loop | 32 |

*Table 22.* Compute estimates for each step of ACTIVEULTRAFEEDBACK, estimated in GPU hours.

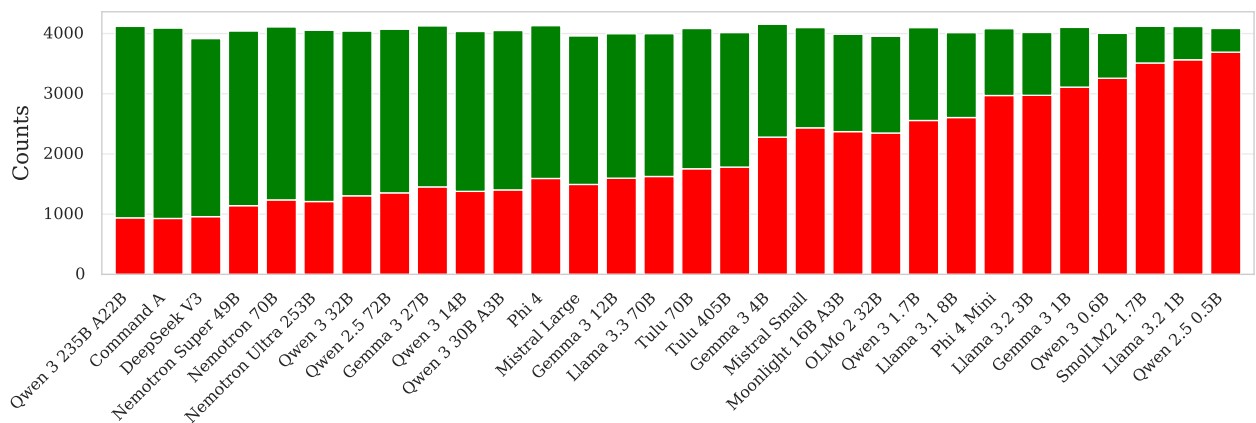

*(a)* **RANDOM**: Model distribution of how often each model in our model pool has been selected by the RANDOM response pair selection method. We further split this data into the number of times each model has been annotated as chosen (green) and rejected (red). Models are sorted based on the number of times they have been annotated as chosen.

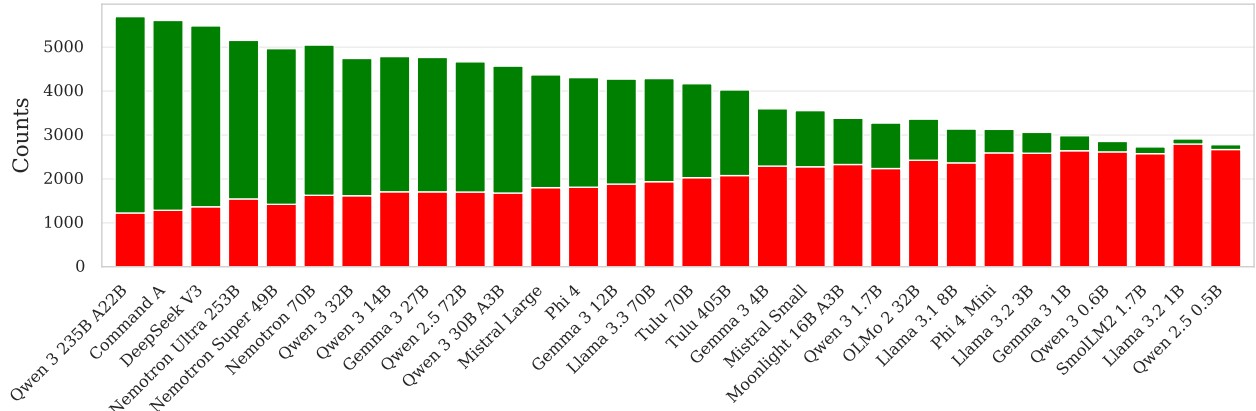

*(b)* **ULTRAFEEDBACK**: Model distribution of how often each model in our model pool has been selected by the ULTRAFEEDBACK response pair selection method. We further split this data into the number of times each model has been annotated as chosen (green) and rejected (red). Models are sorted based on the number of times they have been annotated as chosen.

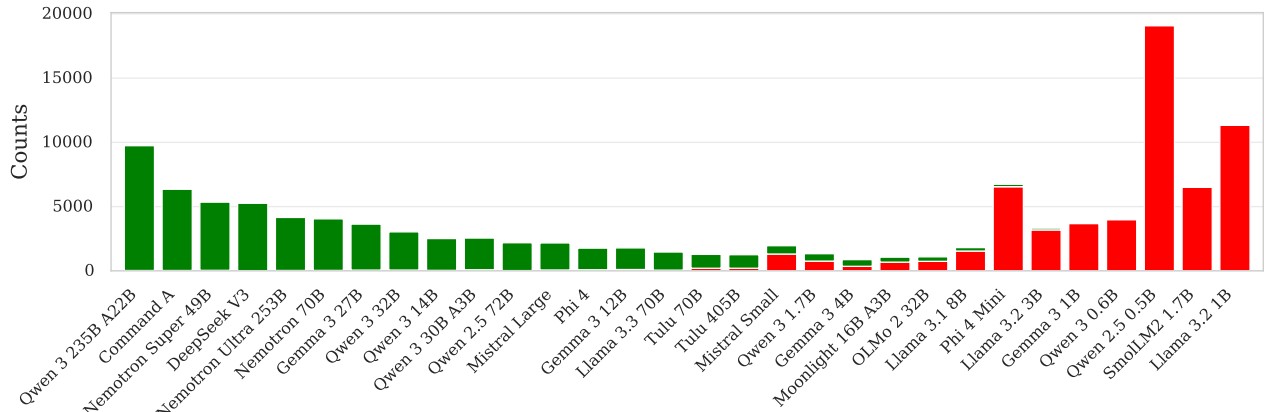

*(c)* **MAXMIN**: Model distribution of how often each model in our model pool has been selected by the MAXMIN response pair selection method. We further split this data into the number of times each model has been annotated as chosen (green) and rejected (red). Models are sorted based on the number of times they have been annotated as chosen.

*Figure 6.* Comparison between the number of times each model from our model pool (Section B.1) has been selected as chosen and rejected model on the **UltraFeedback** prompts for all response pair selection methods we consider.

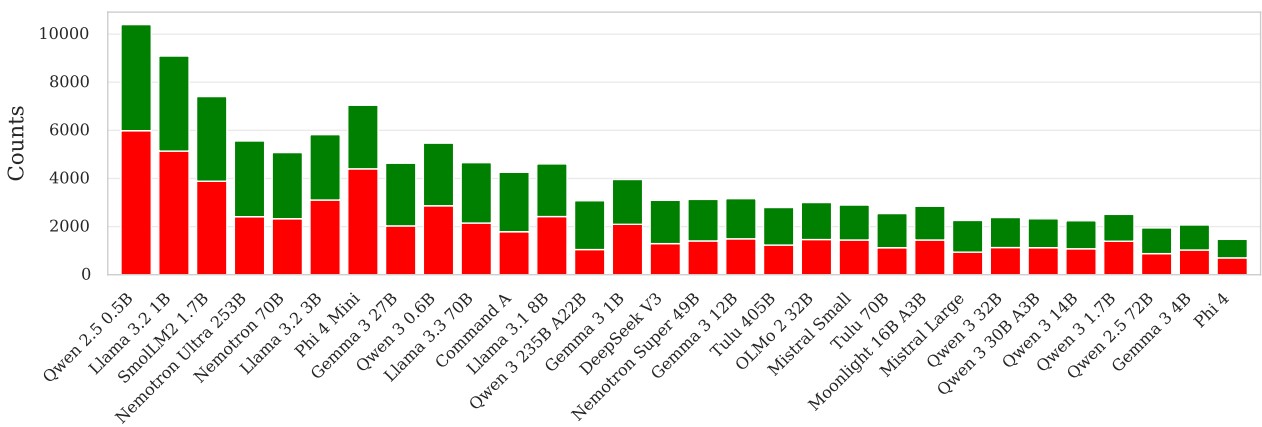

*(d)* **INFOMAX**: Model distribution of how often each model in our model pool has been selected by the INFOMAX response pair selection method. We further split this data into the number of times each model has been annotated as chosen (green) and rejected (red). Models are sorted based on the number of times they have been annotated as chosen.

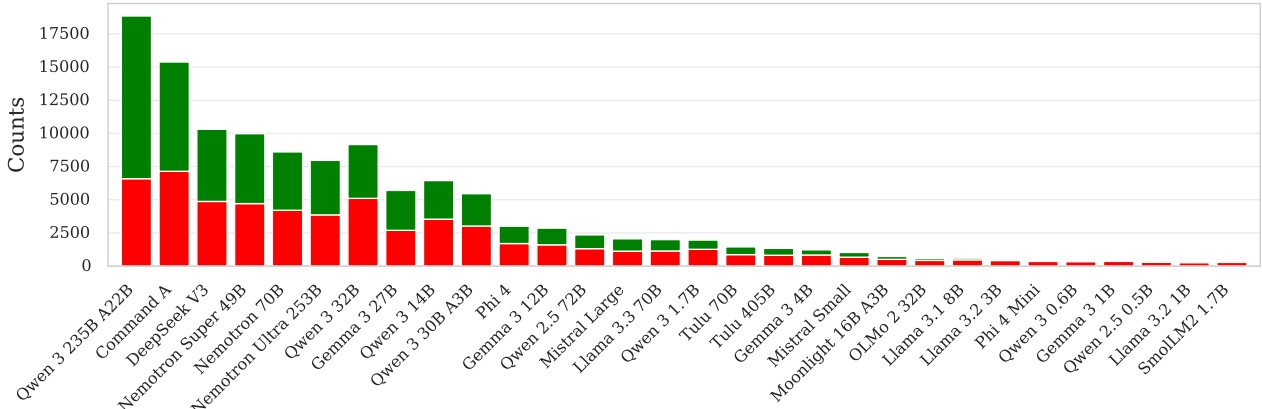

*(e)* **DTS**: Model distribution of how often each model in our model pool has been selected by the DTS response pair selection method. We further split this data into the number of times each model has been annotated as chosen (green) and rejected (red). Models are sorted based on the number of times they have been annotated as chosen.

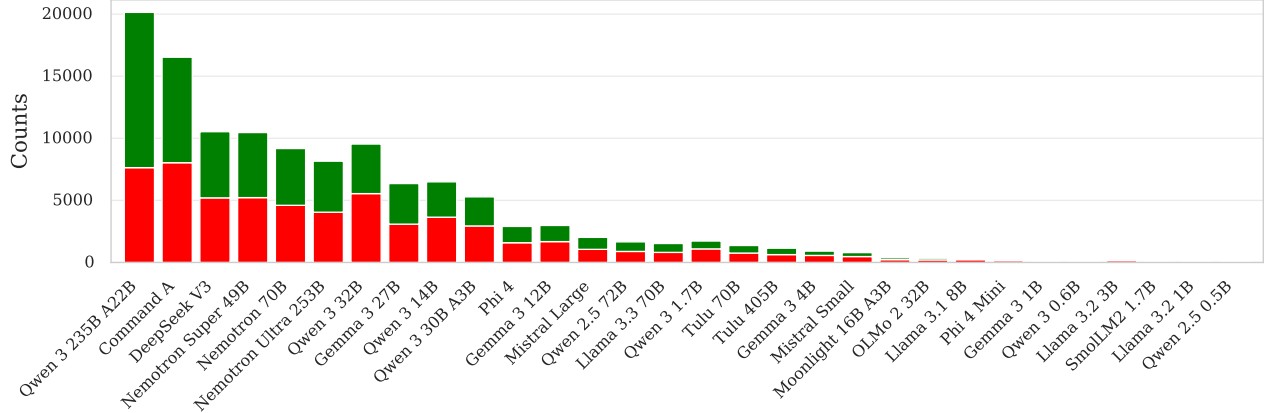

*(f)* **MAXMINLCB**: Model distribution of how often each model in our model pool has been selected by the MAXMINLCB response pair selection method. We further split this data into the number of times each model has been annotated as chosen (green) and rejected (red). Models are sorted based on the number of times they have been annotated as chosen.

*Figure 6.* Comparison between the number of times each model from our model pool (Section B.1) has been selected as chosen and rejected model on the **UltraFeedback** prompts for all response pair selection methods we consider.

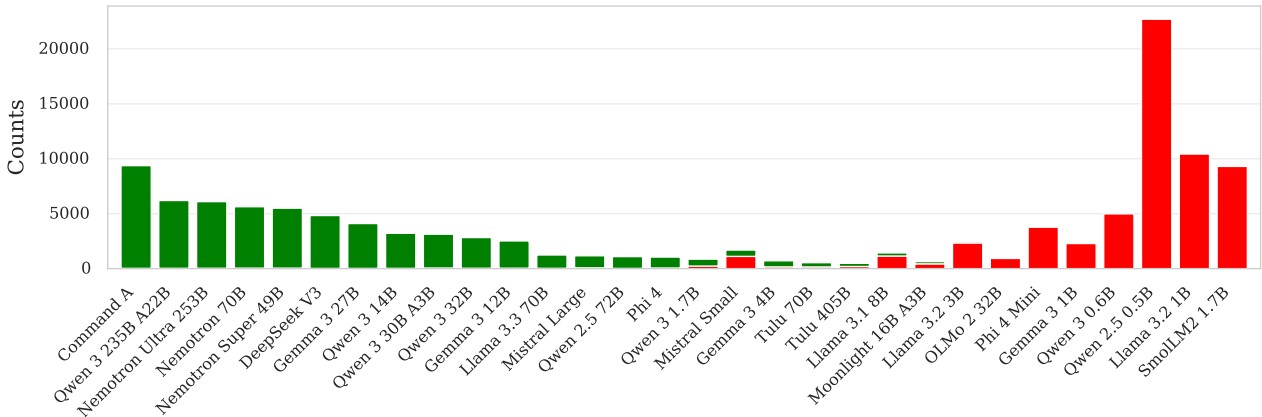

*(g)* **DRTS**: Model distribution of how often each model in our model pool has been selected by the DRTS response pair selection method. We further split this data into the number of times each model has been annotated as chosen (green) and rejected (red). Models are sorted based on the number of times they have been annotated as chosen.

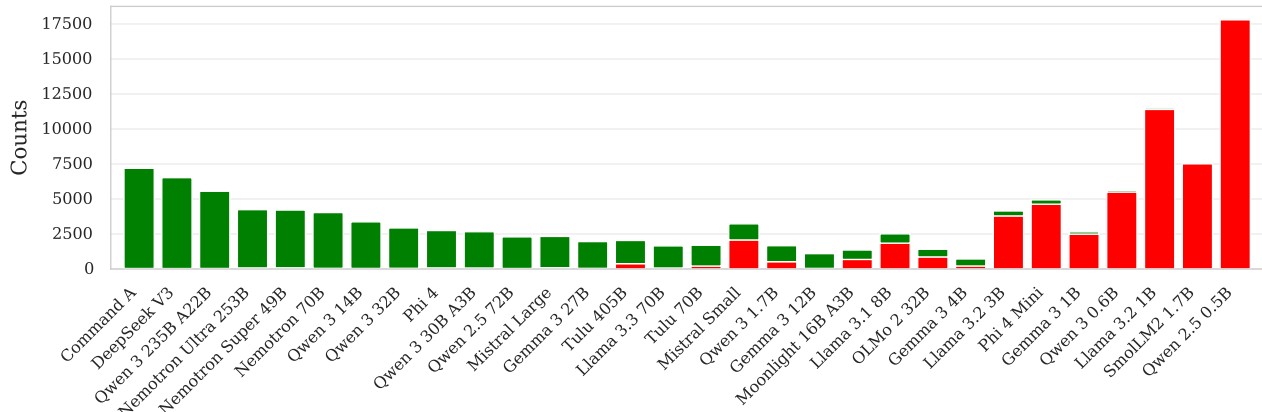

*(h)* **DELTAUCB**: Model distribution of how often each model in our model pool has been selected by the DELTAUCB response pair selection method. We further split this data into the number of times each model has been annotated as chosen (green) and rejected (red). Models are sorted based on the number of times they have been annotated as chosen.

*Figure 6.* Comparison between the number of times each model from our model pool (Section B.1) has been selected as chosen and rejected model on the **UltraFeedback** prompts for all response pair selection methods we consider.

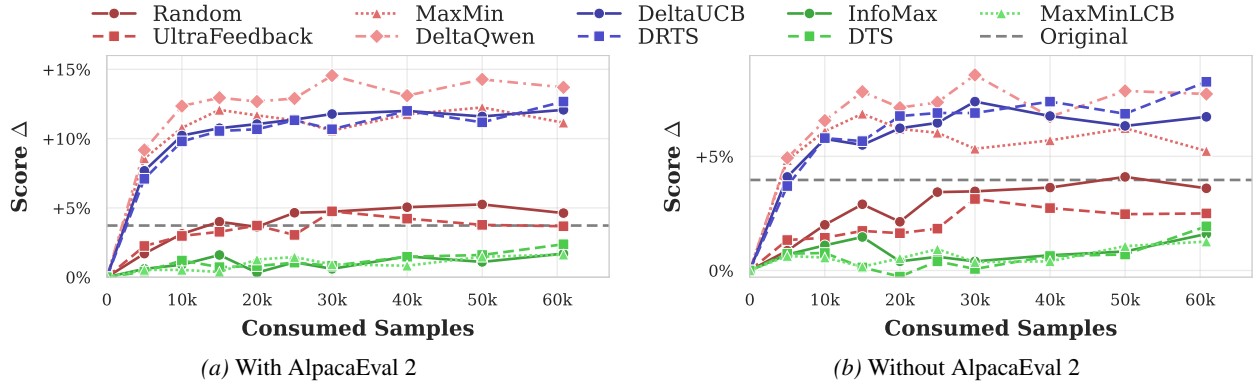

*(a)* With AlpacaEval 2

*(b)* Without AlpacaEval 2

*Figure 7.* Results for the sample efficiency experiment (Section 5.3). We compare the aggregate scores with and without AlpacaEval 2 to demonstrate how its larger magnitude dominates the mean across all benchmarks.

*Table 24.* Results of ACTIVEULTRAFEEDBACK on downstream and reward model benchmarks using different prompt input datasets and response pair selection methods. All scores are given as relative deltas to the base model's scores for readability. Best scores are in bold. We furthermore show the scores obtained by training on the actual **UltraFeedback**, **Skywork**, and **Tulu 3** preference mixture datasets.

| Method | GSM8K | IFEval | TruthfulQA | AlpacaEval 2 | Mean | RewardBench 2 |
|---|---|---|---|---|---|---|
| Base Model | 0.758 | 0.713 | 0.468 | 0.083 | 0.506 | 0.290 |
| **UltraFeedback Prompts** | | | | | | |
| Original | +0.039 | +0.025 | +0.055 | +0.030 | +0.037 | +0.295 |
| RANDOM | +0.024 | +0.028 | +0.056 | +0.077 | +0.046 | +0.278 |
| ULTRAFEEDBACK | +0.037 | -0.001 | +0.039 | +0.072 | +0.036 | +0.287 |
| MAXMIN | +0.022 | -0.016 | **+0.150** | +0.289 | +0.111 | +0.318 |
| DELTAQWEN | **+0.055** | +0.047 | +0.130 | **+0.316** | **+0.137** | +0.100 |
| INFOMAX | +0.011 | +0.019 | +0.018 | +0.020 | +0.016 | +0.297 |
| DTS | +0.011 | +0.034 | +0.013 | +0.037 | +0.023 | +0.224 |
| MAXMINLCB | +0.015 | +0.017 | +0.006 | +0.027 | +0.016 | +0.230 |
| DRTS | **+0.055** | **+0.050** | +0.143 | +0.259 | +0.127 | +0.312 |
| DELTAUCB | +0.040 | +0.025 | +0.137 | +0.281 | +0.120 | **+0.339** |
| **Skywork Prompts** | | | | | | |
| Original | +0.008 | +0.052 | +0.048 | +0.066 | +0.044 | **+0.377** |
| RANDOM | +0.012 | +0.015 | +0.045 | +0.063 | +0.033 | +0.223 |
| ULTRAFEEDBACK | +0.027 | **+0.054** | +0.043 | +0.071 | +0.048 | +0.234 |
| MAXMIN | +0.049 | -0.011 | +0.128 | +0.270 | +0.108 | +0.325 |
| DELTAQWEN | **+0.058** | +0.002 | **+0.152** | **+0.384** | **+0.149** | +0.129 |
| INFOMAX | +0.021 | +0.002 | +0.011 | +0.013 | +0.012 | +0.244 |
| DTS | +0.008 | +0.002 | +0.011 | +0.021 | +0.010 | +0.219 |
| MAXMINLCB | +0.003 | +0.010 | +0.004 | +0.018 | +0.008 | +0.184 |
| DRTS | +0.052 | +0.012 | +0.114 | +0.229 | +0.101 | +0.256 |
| DELTAUCB | +0.055 | +0.013 | +0.077 | +0.238 | +0.095 | +0.262 |
| **Combined Prompts** | | | | | | |
| Original | +0.035 | **+0.049** | +0.051 | +0.030 | +0.041 | **+0.378** |
| RANDOM | +0.043 | +0.012 | +0.074 | +0.036 | +0.041 | +0.269 |
| ULTRAFEEDBACK | +0.043 | +0.032 | +0.056 | +0.086 | +0.054 | +0.240 |
| MAXMIN | +0.027 | +0.023 | **+0.149** | +0.304 | +0.125 | +0.325 |
| DELTAQWEN | +0.048 | +0.000 | **+0.149** | **+0.386** | **+0.145** | +0.153 |
| INFOMAX | +0.011 | +0.021 | +0.014 | +0.018 | +0.015 | +0.300 |
| DTS | +0.009 | +0.002 | +0.014 | +0.029 | +0.013 | +0.247 |
| MAXMINLCB | -0.010 | +0.019 | +0.010 | +0.021 | +0.009 | +0.219 |
| DRTS | **+0.055** | +0.015 | +0.108 | +0.177 | +0.088 | +0.309 |
| DELTAUCB | +0.049 | +0.039 | +0.117 | +0.217 | +0.105 | +0.292 |
| **Tulu 3 Prompts** | | | | | | |
| Original | +0.037 | **+0.069** | +0.046 | +0.020 | +0.043 | +0.297 |
| RANDOM | **+0.055** | +0.041 | +0.069 | +0.046 | +0.052 | +0.360 |
| ULTRAFEEDBACK | +0.043 | +0.052 | +0.056 | +0.057 | +0.051 | +0.343 |
| MAXMIN | +0.022 | +0.067 | **+0.188** | +0.279 | **+0.138** | +0.344 |
| DELTAQWEN | +0.049 | +0.034 | +0.124 | **+0.291** | +0.124 | +0.085 |
| INFOMAX | +0.021 | +0.008 | +0.039 | +0.012 | +0.020 | +0.306 |
| DTS | +0.015 | +0.012 | +0.018 | +0.024 | +0.017 | +0.243 |
| MAXMINLCB | +0.013 | -0.014 | +0.012 | +0.019 | +0.008 | +0.264 |
| DRTS | +0.050 | +0.058 | +0.118 | +0.203 | +0.107 | +0.348 |
| DELTAUCB | +0.028 | +0.060 | +0.134 | +0.235 | +0.114 | **+0.383** |

*Table 25.* Results of ACTIVEULTRAFEEDBACK on downstream benchmarks using different preference tuning algorithms and response pair selection methods. All scores are given as relative deltas to the base model's scores for readability. Best score highlighted in bold.

| Algorithm | Method | GSM8K | IFEval | TruthfulQA | AlpacaEval 2 | Mean |
|---|---|---|---|---|---|---|
| – | Base Model | 0.758 | 0.713 | 0.468 | 0.083 | 0.506 |
| DPO | RANDOM | +0.024 | +0.028 | +0.056 | +0.077 | +0.046 |
| | ULTRAFEEDBACK | +0.037 | -0.001 | +0.039 | +0.072 | +0.036 |
| | MAXMIN | +0.022 | -0.016 | **+0.150** | +0.289 | +0.111 |
| | DELTAQWEN | **+0.055** | +0.047 | +0.130 | **+0.316** | **+0.137** |
| | INFOMAX | +0.011 | +0.019 | +0.018 | +0.020 | +0.016 |
| | DTS | +0.011 | +0.034 | +0.013 | +0.037 | +0.023 |
| | MAXMINLCB | +0.015 | +0.017 | +0.006 | +0.027 | +0.016 |
| | DRTS | **+0.055** | **+0.050** | +0.143 | +0.259 | +0.127 |
| | DELTAUCB | +0.040 | +0.025 | +0.137 | +0.281 | +0.120 |
| IPO | RANDOM | +0.066 | -0.099 | +0.113 | +0.415 | +0.123 |
| | ULTRAFEEDBACK | **+0.074** | +0.000 | +0.050 | +0.415 | +0.135 |
| | MAXMIN | +0.069 | -0.007 | **+0.127** | +0.416 | +0.151 |
| | DELTAQWEN | +0.057 | **+0.039** | +0.025 | +0.275 | +0.098 |
| | INFOMAX | -0.757 | -0.312 | +0.097 | -0.082 | -0.264 |
| | DTS | +0.059 | -0.070 | +0.046 | **+0.480** | +0.128 |
| | MAXMINLCB | +0.005 | +0.013 | -0.002 | +0.013 | +0.007 |
| | DRTS | +0.051 | +0.030 | +0.111 | +0.441 | **+0.158** |
| | DELTAUCB | +0.060 | +0.010 | +0.101 | +0.333 | +0.126 |
| SimPO | RANDOM | +0.046 | -0.007 | +0.133 | +0.496 | +0.166 |
| | ULTRAFEEDBACK | +0.038 | -0.042 | +0.163 | **+0.568** | **+0.181** |
| | MAXMIN | +0.007 | -0.059 | **+0.185** | +0.460 | +0.148 |
| | DELTAQWEN | **+0.063** | **+0.019** | +0.065 | +0.435 | +0.145 |
| | INFOMAX | -0.004 | -0.024 | +0.042 | +0.037 | +0.013 |
| | DTS | -0.058 | -0.147 | +0.083 | +0.536 | +0.103 |
| | MAXMINLCB | -0.006 | -0.022 | +0.038 | +0.020 | +0.007 |
| | DRTS | +0.054 | -0.005 | +0.162 | +0.514 | **+0.181** |
| | DELTAUCB | +0.044 | -0.029 | +0.177 | +0.509 | +0.175 |

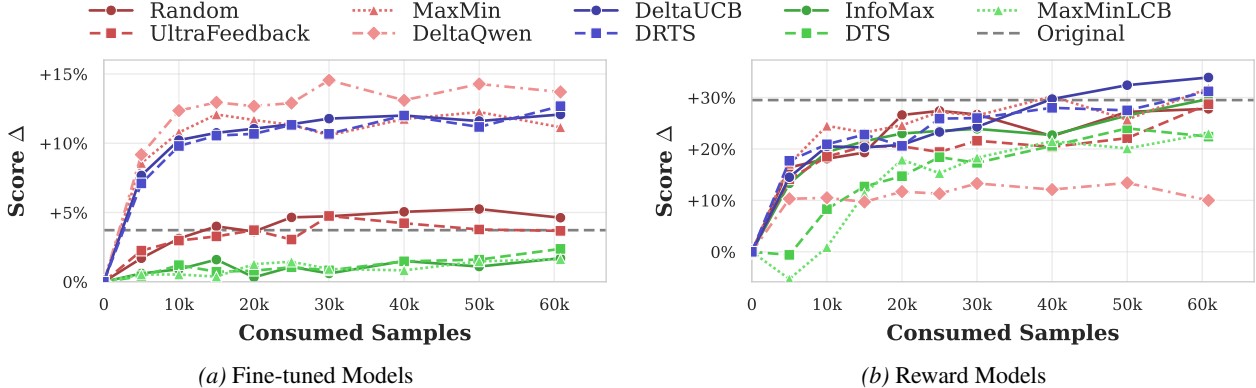

*(a)* Fine-tuned Models      *(b)* Reward Models

*Figure 8.* Mean performance trajectories for fine-tuned and reward models as a function of consumed samples on the ACTIVEULTRA-FEEDBACK prompt pool using DPO. All curves share the same prompts and differ only in the response pair selection strategy. We compare datasets generated via ACTIVEULTRAFEEDBACK using various selection methods, and also report the score achieved by the original **UltraFeedback** dataset (Cui et al., 2024) with its original response pairs.

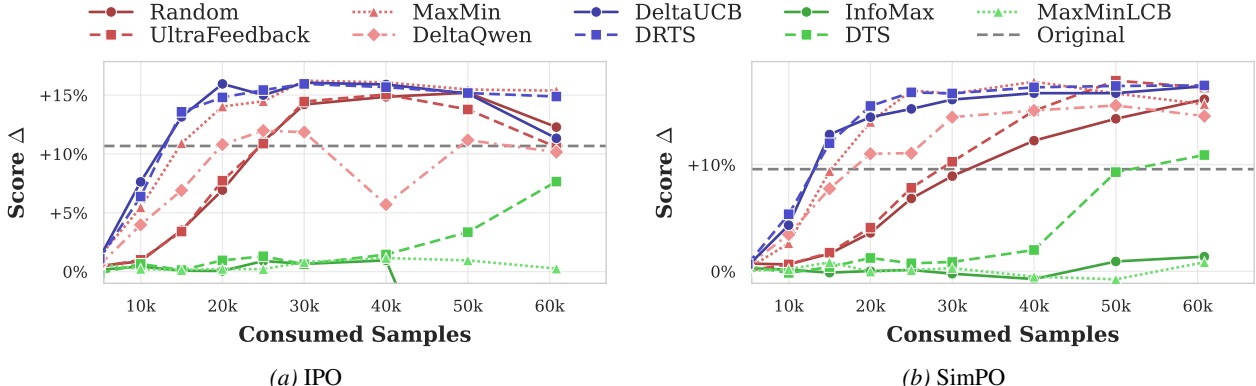

*Figure 9.* Mean performance trajectories for our fine-tuned models using IPO (Figure 9a) and SimPO (Figure 9b) as a function of consumed samples on datasets generated using ACTIVEULTRAFEEDBACK based on **UltraFeedback** prompts. We provide the scores achieved using the original preference dataset instead of just the prompts with ACTIVEULTRAFEEDBACK for reference.

*Table 26.* Sensitivity of DRTS to the size and diversity of the response model pool. Scores are reported as deltas relative to the base model after DPO training.

| Model Pool Composition | GSM8K | IFEval | TruthfulQA | AlpacaEval 2 | Mean |
|---|---|---|---|---|---|
| All 30 models | +0.060 | +0.019 | +0.129 | +0.240 | +0.112 |
| 15 largest models only | +0.011 | -0.001 | -0.003 | +0.059 | +0.017 |
| 15 smallest models only | +0.050 | +0.036 | +0.100 | +0.105 | +0.073 |
| 5 largest + 5 smallest models | +0.070 | +0.025 | +0.114 | +0.152 | +0.090 |

