# OpenReview forum: "ActiveUltraFeedback: Efficient Preference Data Generation using Active Learning"
_ICML.cc/2026/Conference — ICML 2026 regular_

### Official Review · Reviewer_iSKm · 2026-03-05

**Soundness:** 3
**Presentation:** 2
**Significance:** 2
**Originality:** 2
**Overall Recommendation:** 4
**Confidence:** 3

**Summary:**

The article's major idea is to improve preference data generation for RLHF via an active learning pipeline. Overall, the authors address a central concept in RLHF data collection: selecting informative response pairs for annotation. The proposed ActiveUltraFeedback framework estimates reward uncertainty and introduces two pair selection strategies, DRTS and DeltaUCB. Experimental results indicate improved sample efficiency compared with existing heuristic methods.

**Compliance With Llm Reviewing Policy:**

Affirmed.

**Key Questions For Authors:**

1. The pipeline reduces annotation cost but requires generating responses from many LLMs. Can the authors quantify the overall compute cost compared to static pipelines such as UltraFeedback?
2. The preference labels rely entirely on an LLM judge. Have the authors evaluated how well these labels correlate with human preferences?
3. How sensitive are the results to the diversity and size of the response model pool?
4. Can the authors provide additional analysis explaining why delta-based response selection improves dataset quality?

**Limitations:**

Refer to the key question

**Strengths And Weaknesses:**

Strength
1.The paper addresses an important problem in RLHF, namely improving the efficiency of preference data collection. Reducing annotation cost is highly relevant for scaling alignment pipelines.
2.The proposed framework is modular and flexible, allowing different response generation models, reward estimators, and pair selection strategies to be integrated.
3.Comprehensive evaluations across datasets, algorithms, and tasks, plus stability analyses (random seeds, system non-determinism), ensure reliable results.

Weakness
1.The pipeline relies on generating responses from a large pool of LLMs, which introduces significant computational overhead that is not thoroughly analyzed.
2.Preference labels are generated using an LLM judge rather than human annotations, raising concerns about potential bias and evaluation circularity.

---

> ### Author Rebuttal · Authors · 2026-03-31
>
> Dear Reviewer iSKm,
>
> Thank you for reviewing our submission and the constructive feedback. We appreciate that you recognize our contribution of reducing expensive human annotation for preference data generation and our framework's modularity, which we designed for future work to easily build upon and adapt.
>
> We respond to your questions and concerns below.
>
> 1. **On the efficiency of ActiveUltraFeedback (Q1)**
>
>     The main focus of our work is annotation efficiency, given how human annotation costs around \\$10 per sample \[1\] whereas response generation costs only cents \[2\].
>
>     As we show in Figures 3 and 5, DRTS and DeltaUCB can achieve similar or even superior performance with 3-6 times fewer samples than the UltraFeedback selection method. In the case of DPO fine-tuning depicted in Figure 3, UltraFeedback selection reaches peak performance after 30,000 samples. Creating this dataset involves generating and annotating four responses per prompt, which would cost \\$24,000 with an LLM annotator. In contrast, DRTS matches this performance with just 5,000 samples. Producing this dataset involves generating 30 responses but annotating only two per prompt, which costs just \\$16,000 with an LLM annotator and accounts for a 1.5x reduction. The cost saving is even more pronounced with human annotators: assuming they are tasked to annotate multiple responses simultaneously, the UltraFeedback dataset would cost \\$312,000 whereas the DRTS dataset would cost just \$65,000, which is almost 5x less.
>
>     We would also like to clarify that response generation consumed 600 GPU hours because we generated the 30 responses for all 60,000 prompts in advance to facilitate large-scale experimentation. In practice, using our pipeline with a strong selection method such as DRTS or DeltaUCB would require significantly fewer samples and thus significantly less response generation. For further efficiency, we find that competitive performance with DRTS can still be achieved using a high-diversity pool of just 15 models, as outlined in our response to Question 3.
>
> 2. **Using LLM-as-a-judge to simulate human annotations (Q2)**
>
>     We use LLM-as-a-judge to pursue our goal of reducing human annotation cost, and we validate this choice by showing high correlation with human annotations on AlpacaEval 2.0 and strong performance on RewardBench 2. For a detailed discussion, we refer to our response to Reviewer VmuH, Question 2.
>
> 3. **Sensitivity of results to diversity and size of response model pool (Q3)**
>
>     To investigate the importance of model pool diversity and size, we ran DRTS on four modified model pools categorized by parameter count. The results on downstream tasks after DPO training on each dataset are in the table below.
>
>     | Model Pool Composition | GSM8K | IFEval | TruthfulQA | AlpacaEval 2 | Mean |
>     | - | - | - | - | - | - |
>     | All 30 models | +0.060 | +0.019 | +0.129 | +0.240 | +0.112 |
>     | 15 largest models only | +0.011 | -0.001 | -0.003 | +0.059 | +0.017 |
>     | 15 smallest models only | +0.050 | +0.036 | +0.100 | +0.105 | +0.073 |
>     | 5 largest models + 5 smallest models | +0.070 | +0.025 | +0.114 | +0.152 | +0.090 |
>
>     The 15 largest-only pool performs significantly worse (Mean +0.017), highlighting the importance of diversity. The 15 smallest-only pool performs reasonably well (Mean +0.073) because the largest model is 24B parameters, providing some quality delta. Crucially, the highly-diverse pool of 5 largest and 5 smallest models achieves competitive performance (Mean +0.090), further showing that the number of generations required can be reduced while maintaining quality.
>
> 4. **Analysis on why delta-based response selection improves dataset quality (Q4)**
>
>     We would like to note that delta-based methods were introduced by Geng et al. (2025) and have shown great success in open-source LLMs such as SmolLM 3 [Bakouch et al., 2025] and Olmo 3 [Olmo et al., 2025]. We provide an explanation based on the analysis in Section 6 of Geng et al. (2025):
>
>     A clear quality delta between the chosen and rejected responses isolates the distinct features that make one better than the other. This provides a strong, low-variance gradient signal that points the model in the direction of improvement within the reward landscape, rather than simply encouraging it to clone the text distribution of the chosen response. By prioritizing pairs with these highly informative deltas, delta-learning ensures robust directional updates and avoids the noisy, inefficient learning that occurs when response pairs have ambiguous preference boundaries.
>
> We hope we have addressed your concerns and would be grateful if you could consider adjusting your score accordingly. We are happy to engage further if needed.
>
> \[1\]: https://docs.google.com/presentation/d/179dpzWSQ9G7EAUlvaJdeE0av9PLuk9Rl33nfhHSJ4xI/edit?slide=id.g31d042f750e_1_0#slide=id.g31d042f750e_1_0
>
> \[2\]: https://epoch.ai/data-insights/llm-inference-price-trends

---

> > ### Author Rebuttal · Reviewer_iSKm · 2026-04-01
> >
> > Thank you for your reply, you have addressed all my concerns, and I will raise the score.

---

### Official Review · Reviewer_VmuH · 2026-03-10

**Soundness:** 3
**Presentation:** 2
**Significance:** 3
**Originality:** 2
**Overall Recommendation:** 3
**Confidence:** 4

**Summary:**

This paper explores how to efficiently generate preference datasets required for Reinforcement Learning from Human Feedback (RLHF) through Active Learning. Traditional preference data generation relies on static heuristics, which often leads to inefficient sampling and consumes substantial annotation costs. To address this, the authors propose the ActiveUltraFeedback framework, modeling response selection as a Contextual Dueling Bandit problem. This framework employs an Ensemble Neural Network (ENN) based on ensemble multilayer perceptrons to estimate rewards and their epistemic uncertainty. Based on the Delta Learning Hypothesis, the authors argue that traditional active learning methods aimed at minimizing regret are unsuitable for preference data generation. Accordingly, the paper proposes two novel response pair selection algorithms: Dual Reverse Thompson Sampling (DRTS) and DeltaUCB. These methods prioritize selecting response pairs with the largest predicted quality gap, rather than solely pursuing high-quality responses. Experiments demonstrate that preference datasets generated using DRTS and DeltaUCB significantly enhance performance in both reward model training and downstream policy fine-tuning (covering algorithms like DPO, IPO, and SimPO). Their most prominent contribution lies in exceptional sample efficiency: achieving equivalent or superior downstream evaluation performance with only one-third to one-sixth the annotation volume required by traditional static benchmark datasets.

**Compliance With Llm Reviewing Policy:**

Affirmed.

**Final Justification:**

The authors provided a rebuttal that resolved my specific technical queries regarding annotation efficiency, LLM-as-a-judge validation and ENN design choices. However, having carefully re-evaluated the manuscript alongside the new clarifications, I have decided to maintain my original score. While I recognise the paper as an empirical study with extensive cross-task evaluations, its core contribution remains an incremental combination of existing pipelines rather than a fundamental algorithmic breakthrough.

**Key Questions For Authors:**

1. Your framework needs to generate a massive number of candidate responses using a large model pool of m=30 before each round of evaluation. In real-world applications, the computational cost of inference generation itself is a significant bottleneck. Considering both "candidate generation cost" and "judge annotation cost," how does the overall computational efficiency of ActiveUltraFeedback compare to a static baseline that generates only 2-4 candidates but annotates them all (such as the original UltraFeedback process)?

2. In Appendix D.1, probabilistic scoring suppresses the inference process and reduces the tie rate to an astonishing 0.0%. However, forcing the judge model to produce numerical differences between two potentially extremely similar or equally good responses raises the question of whether the model will create an "illusion" of poor quality based on spurious features unrelated to semantic quality (such as punctuation marks or minor length differences). Have you validated the reliability of this continuous scoring on datasets with high-quality human annotations (e.g., containing true tie labels)?

3. The current ENN architecture uses a frozen 4B model backbone network with 20 independent MLP prediction heads attached. Given that the active learning loop needs to compute the uncertainty of all candidate pairs in each batch, will the performance of DRTS and DeltaUCB deteriorate sharply if a smaller ensemble size or a lighter backbone is adopted in future deployments to reduce inference latency? How tolerant are they to the accuracy of uncertainty estimation?

**Limitations:**

Yes

**Strengths And Weaknesses:**

Soundness:

Strengths: The experimental design is rigorous and systematic. The authors not only performed thorough ablation on multiple cue datasets of varying sizes (UltraFeedback, Skywork, Tulu 3), but also validated the data's generalization ability across various preference optimization algorithms (DPO, IPO, SimPO). More commendably, the authors decoupled the evaluation of "reward model training" and "policy fine-tuning," revealing a key flaw where certain static baselines (such as DeltaQwen) perform well in fine-tuning but generalize extremely poorly in reward modeling. Stability analyses of hyperparameters, random seeds, and system-level nondeterminism in the appendix (Appendix E.2) further strengthen the technical reliability of the conclusions.

Weaknesses: The entire evaluation loop heavily relies on LLM as the referee to replace human annotations. While this is feasible when scaling up experiments, it completely lacks direct alignment validation with real human preferences. In particular, while its "probabilistic scoring" method eliminates ties (0.0% tie rate), it hasn't been proven whether this small probability difference calculated by extracting logits truly reflects differences in semantic quality perceived by humans, or whether it amplifies the inherent numerical bias of a particular referee model.

Presentation:
Weaknesses: The visualization of core results (such as Figures 3 and 5) is somewhat overcrowded because it includes as many as 10 line graphs of the comparative baselines. Without magnification, it is difficult for readers to quickly distinguish the performance of methods in the middle tier. It is recommended to highlight the best-performing methods (such as DRTS, DeltaUCB) and core comparison baselines (such as Random, Original) in the main text charts, while moving the complete full-scale line graphs to the appendix.

Significance:
Strengths: Data efficiency is a key pain point in the current large language model alignment stage of RLHF. This paper empirically challenges an important intuition: traditional dueling methods (such as DTS, MaxMinLCB) tend to select two good but minimally different responses, resulting in generated preference data lacking sufficient "gradient" signals, thus performing worse than random sampling.

Weaknesses: The paper emphasizes high "efficiency," but this is mainly reflected in annotation efficiency. To run this Active Learning loop, the system must maintain a large model pool containing 30 models at the front end and generate 30 candidate responses for each Prompt (the authors estimate that the generation step alone consumes approximately 600 GPU hours). In real-world industrial pipelines, the computational cost of generating such massive amounts of data is extremely high, potentially offsetting the benefits of reducing API annotation counts.

Originality：
Strengths: The most significant highlight of this paper is the ingenious combination of uncertainty estimation in active learning with the "incremental learning hypothesis."

Weaknesses: Applying active learning (especially uncertainty estimation based on ENNs) to large-scale model preference alignment is not original (e.g., works cited by the authors such as Dwaracherla et al., 2024; Liu et al., 2024c). The originality of this paper is primarily limited to the innovative combination of specific acquisition functions and the first large-scale deployment and cross-task (RM and DPO) system evaluation.

---

> ### Author Rebuttal · Authors · 2026-03-31
>
> Dear Reviewer VmuH,
>
> Thank you for reviewing our submission and giving constructive feedback. We appreciate your recognition of our contribution to reducing expensive human annotation, our extensive experiments validating our approach, and our findings on the underperformance of regret minimization-based response selection.
>
> We respond to your questions and concerns below.
>
> 1. **On the efficiency of ActiveUltraFeedback (Q1)**
>
>     Our focus is on annotation efficiency, the most expensive part of data collection. Our gains in this regard are significant: we achieve outstanding performance while requiring far fewer annotations, yielding a 3-6x cost reduction. For a detailed discussion, we refer to the first question in our rebuttal to Reviewer iSKm.
>
> 2. **Using LLM-as-a-judge to simulate human annotations (Q2)**
>
>     Our contribution focuses on selecting informative pairs for LLM fine-tuning, not simulating human preferences. Our main goal is reducing the need for human annotation, which is the most expensive part of preference data collection, but is necessary in low-resource and expert domains where current LLMs cannot yet provide reliable feedback. We use LLM annotations to facilitate experiments at scale.
>
>     Nevertheless, we understand that reliable LLM feedback is essential for drawing conclusions from our work. For this reason, we already include evaluations on RewardBench 2 in Table 5, Appendix D.2, which has high correlation with downstream performance [Malik et al., 2025]. We also extended this evaluation to a discrete judge that produces integer scores directly, in line with prior work [Cui et al., 2024], and found a reduction in the mean score from 0.744 to 0.698, and from 0.833 to 0.729 on the Ties subtask, which measures judge sensitivity for similar answers. We further compared the two LLM judge approaches against human annotations on AlpacaEval 2.0 [Dubois et al., 2024], and found that both have similarly high correlation with human annotations, Spearman’s r = 0.67 and 0.70 for our continuous judge and the discrete judge, respectively. However, our continuous judge achieves higher human agreement (62.7% vs 60.4%).
>
>     We see further improvements in this area as a promising avenue for future work, and have added these additional results to the paper.
>
> 3. **On the design choices of the ENN model (Q3)**
>
>     We used Skywork Reward V2 Qwen3 4B as the base model for the ENN due to its strong performance on RewardBench 2. This is a relatively small model, which helps keep inference latency low, and we recommend reducing latency even further by conducting certain steps asynchronously. For instance, as we describe in Appendix E.3, we keep the base model frozen. This means that the last layer embeddings of the base model for one batch can be computed while the previous batch is being annotated. Furthermore, we find training the MLP reward heads in the loop very efficient, as shown in Table 20, Appendix E.4, because they are quite small. Finally, our implementation of the ENN follows prior works [Dwaracherla et al., 2024; Melo et al., 2024; Liu et al., 2024c], which conducted extensive experiments to find hyperparameters that lead to robust uncertainty estimations. We see studying how uncertainty estimation accuracy impacts our pipeline's performance as a promising direction for future work.
>
> 4. **On the originality of our contribution**
>
>     While we build upon the existing pipelines, models such as ENNs, and active learning approaches, we respectfully argue that the combination of these methods and carrying out downstream evaluations is the first-of-its-kind in the literature. Furthermore, our contributions extend beyond combining existing techniques innovatively.
>
>     - We introduce two novel response pair selection methods: DRTS and DeltaUCB, which consistently outperform standard dueling bandit baselines and prior heuristics on various benchmarks and on sample efficiency, and their success generalizes across datasets and optimization algorithms.
>
>     - We want to emphasize that our large-scale, cross-task evaluation is an important scientific contribution that corrects assumptions in the literature. For example, prior work [Dwaracherla et al., 2024] successfully adapted traditional dueling bandit methods like DTS for reward model training. However, we find that these methods do not transfer effectively to the task of preference data generation for model fine-tuning because they yield datasets that lack the quality deltas required for learning, causing them to underperform even random sampling. Uncovering these blind spots is a direct and highly relevant result of our work.
>
> 5. **Visualizations of core results are overcrowded**
>
>     We improved the readability of Figures 3 and 5 by keeping only DRTS, DeltaUCB, UltraFeedback, DeltaQwen, and DTS, and moved the complete plots to the appendix.
>
> We believe we have addressed your questions and would be glad if you could reassess your score accordingly.

---

> > ### Author Rebuttal · Reviewer_VmuH · 2026-04-02
> >
> > I have read the responses carefully, and I appreciate the specific clarifications provided. The authors have successfully addressed my initial questions. Specifically, the explanation regarding the 3-6x cost reduction of ActiveUltraFeedback (Q1) is clear. I also appreciate the additional evaluations using RewardBench 2 and AlpacaEval 2.0 to justify the LLM-as-a-judge setup (Q2), as well as the detailed rationale behind the ENN model design, including the use of Skywork Reward V2 and the asynchronous execution strategy (Q3). The specific technical ambiguities I raised have been fully resolved.
> >
> > However, after comprehensively re-evaluating the manuscript alongside the rebuttal, I have decided to maintain my original score. My initial rating already factored in the strengths of this paper, particularly the extensive cross-task empirical evaluations and the introduction of DRTS/DeltaUCB. While the authors successfully defended their design choices, the core contribution—as the authors themselves acknowledged—primarily lies in the novel combination of existing pipelines rather than fundamental algorithmic breakthroughs. While the current framework is a solid empirical study, it remains an incremental advancement. The rebuttal effectively cleared up my doubts but did not present compelling new evidence that shifts the fundamental nature of the contribution to warrant a higher score.

---

> > > ### Author Response · Authors · 2026-04-08
> > >
> > > We are happy to hear that **our rebuttal has satisfactorily addressed all the Reviewer’s questions and doubts**, and appreciate that the Reviewer sees our work as a solid empirical study that not only introduces **two new response pair sampling methods** that outperform existing ones across multiple settings, but also provides **novel insights** on sample efficiency, the delta learning hypothesis, and the failure of regret minimization in preference dataset generation on downstream performance.
> > >
> > > As we understand, the Reviewer’s main concern is with the originality of our work, as stated in the Acknowledgement:
> > > > “the core contribution […] primarily lies in the novel combination of existing pipelines rather than fundamental algorithmic breakthroughs.”
> > >
> > > We would however like to highlight that the Reviewer Instructions (https://icml.cc/Conferences/2026/ReviewerInstructions ) clearly state that under the criteria for Originality:
> > > > “[…] originality does not necessarily require introducing an entirely new method. **Rather, a work that provides novel insights by evaluating existing methods, or demonstrates improved understanding is also equally valuable.**"
> > >
> > > The Reviewer has repeatedly acknowledged and highlighted our novel insights and improved understanding. For example,
> > > > “Their most prominent contribution lies in **exceptional sample efficiency**: achieving equivalent or superior downstream evaluation performance with only one-third to one-sixth the annotation volume required by traditional static benchmark datasets.”
> > >
> > > >  “More commendably, the authors decoupled the evaluation of ‘reward model training’ and ‘policy fine-tuning,’ **revealing a key flaw** where certain static baselines (such as DeltaQwen) perform well in fine-tuning but generalize extremely poorly in reward modeling.”
> > >
> > > > “**This paper empirically challenges an important intuition**: traditional dueling methods (such as DTS, MaxMinLCB) tend to select two good but minimally different responses, resulting in generated preference data lacking sufficient “gradient” signals, thus performing worse than random sampling.”
> > >
> > > Therefore, we believe the Reviewer’s concern about the lack of originality is not in line with the Reviewer Instructions. We believe that **our algorithmic and empirical contributions are valuable jointly**, and we have therefore presented them alongside each other in this work. We would like to ask the Reviewer to reassess their evaluation based on the official criteria of Originality.

---

### Official Review · Reviewer_JX3U · 2026-03-14

**Soundness:** 3
**Presentation:** 4
**Significance:** 3
**Originality:** 3
**Overall Recommendation:** 4
**Confidence:** 4

**Summary:**

This paper studies preference data generation for RLHF under limited labeling budgets. It introduces ActiveUltraFeedback, a modular active learning pipeline that maintains uncertain reward estimates over a diverse response pool and uses them to select one pair per prompt for annotation. Within this framework, the paper benchmarks passive heuristics and dueling-bandit acquisitions, and proposes DRTS and DeltaUCB, two methods that explicitly prefer large predicted quality gaps. The experiments show that these two methods usually outperform random sampling, UltraFeedback-style heuristics, and standard dueling-bandit baselines on both reward-model and policy metrics, and can reach strong downstream performance with substantially fewer selected pairs.

**Compliance With Llm Reviewing Policy:**

Affirmed.

**Key Questions For Authors:**

see weaknesses

**Limitations:**

yes

**Strengths And Weaknesses:**

##  Strengths
- The paper optimizes the right end objective for this problem: it evaluates both reward model quality and downstream policy quality instead of stopping at uncertainty reduction or pair-classification metrics.
- The empirical study is broader than many active-RLHF papers, with comparisons across passive heuristics, dueling-bandit methods, four prompt datasets, and three preference optimization algorithms.
- A useful insight of the paper is that standard dueling-bandit objectives can identify strong responses yet still produce weak preference datasets because they fail to create sufficiently informative quality gaps.
- The appendix adds meaningful supporting evidence through judge ablations, stability analyses, generated-dataset analysis, and compute estimates, which makes the work easier to scrutinize.

## Weaknesses
I do not have major concerns about the paper, but a few points would benefit from clarification:
- For ENN Training (l. 1279), the paper mentions hyperparameter tuning across DPO fine-tuning and reward modeling. It would be helpful to clarify whether all methods were allocated comparable hyperparameter tuning budgets to ensure a fair comparison.
- The results suggest that DeltaQwen performs well with DPO but less favorably with IPO and SimPO. Do you have any analysis or intuition for this observation?

---

> ### Author Rebuttal · Authors · 2026-03-31
>
> Dear Reviewer JX3U,
>
> Thank you for your time reviewing our work, and your helpful feedback. We are glad that you appreciate our focus on practical, large-scale evaluation, particularly our emphasis on downstream policy performance alongside reward model quality. We are particularly happy to see that you have no major concerns about the paper.
>
> Below, we address your questions and the subsequent changes to our submission:
>
> 1. **Hyperparameter tuning budgets across response pair selection methods (Q1)**
>
>     We can confirm that all relevant methods were tuned using the same tuning budget using the grid in Table 16. However, it is important to note that these are Epistemic Neural Network (ENN) specific hyperparameters. The static baseline methods (Random, UltraFeedback, MaxMin, DeltaQwen) do not utilize the ENN; therefore, there was no need to tune the ENN hyperparameters for these methods. For the remaining dueling bandit methods and our novel methods, the tuning budget was the same.
>     We have added this clarification to the section.
>
> 2. **Why DeltaQwen performs well with DPO but less favorably with IPO and SimPO (Q2)**
>
>     We attribute this result to the insufficient diversity of datasets generated using DeltaQwen and its implications on the objective functions and subsequent gradients of each algorithm. Since DeltaQwen is confined to two fixed models, where the smaller model (Qwen 3 0.6B) is also distilled from the bigger model (Qwen 3 32B), the resulting dataset contains similar responses. In comparison, DRTS and DeltaUCB use the full model pool to select different model pairs for each prompt resulting in a higher diversity as shown in Appendix F1, Figure 6.
>
> \\begin{align*}
> \\Delta_{\\theta}(x, y^+, y^-) &= \\log \\frac{\\pi_{\\theta}(y^+ \\mid x)}{\\pi_{\\text{ref}}(y^+ \\mid x)} - \\log \\frac{\\pi_{\\theta}(y^- \\mid x)}{\\pi_{\\text{ref}}(y^- \\mid x)} \\\\
> \\mathcal{L}\_{\\text{DPO}} &= -\\log \\sigma(\\beta \\Delta\_{\\theta}(x, y^+, y^-)) \\\\
> \\mathcal{L}\_{\\text{IPO}} &= \\left( \\Delta\_{\\theta}(x, y^+, y^-) - \\frac{1}{2\\tau} \\right)^2 \\\\
> \\mathcal{L}\_{\\text{SimPO}} &= -\\log \\sigma \\left( \\frac{\\beta}{|y^+|} \\log \\pi\_{\\theta}(y^+ \\mid x) - \\frac{\\beta}{|y^-|} \\log \\pi\_{\\theta}(y^- \\mid x) - \\gamma \\right)
> \\end{align*}
>
>    -  **SimPO**: As mentioned by SimPO [Meng et al., 2024] in their "Preventing catastrophic forgetting without KL regularization" section, when using SimPO, a diverse dataset is a necessary factor to prevent the model from drifting too far away from the reference policy, which would result in catastrophic forgetting. As mentioned above, DeltaQwen can not provide sufficient diversity. This, in combination with SimPO removing the explicit KL divergence regularization from DPO in favor of length normalization, causes the model to drift further, which makes it more prone to catastrophic forgetting and subsequently lower benchmark scores. We confirmed this by checking our training logs, where we found that runs on DeltaQwen indeed end up having a higher KL divergence than those on DRTS and DeltaUCB.
>
>    -  **IPO**: We attribute DeltaQwen's lower performance when using IPO compared to DPO to differences in their objective functions. As established above, DeltaQwen’s lack of diversity tends to cause drift far away from the reference policy, overfitting on the Qwen comparison. In the case of DPO, the gradient will quickly vanish due to the sigmoid function saturating, preventing more extreme overfitting. In comparison, the mean squared error of the IPO objective does not cause vanishing gradients, allowing the model to drift more, which runs into the risk of catastrophic forgetting and subsequently lower benchmark scores, similar to SimPO. We confirmed this by checking our training logs again, which showed higher KL divergence for DeltaQwen in comparison to DRTS, and that gradient norms for DPO on DeltaQwen datasets and for IPO using DRTS were slightly reduced during training, while the gradient norms for IPO using DeltaQwen remained largely constant.
>
>    -  We want to thank you for bringing up this matter, as we believe this could be of interest to other readers as well, and we subsequently extended our "5.5. Preference Optimization Algorithm Ablation" section accordingly.
>
> We hope that our clarifications regarding the ENN hyperparameter setup and the new DeltaQwen intuition address the points you raised. We believe this improved significantly the quality of the paper and we are happy to answer any further questions.

---

> > ### Author Rebuttal · Reviewer_JX3U · 2026-04-04
> >
> > Thank you to the authors for their detailed rebuttal. After careful consideration, I decided to maintain my initial score.

---

### Official Review · Reviewer_YpJu · 2026-03-15

**Soundness:** 3
**Presentation:** 3
**Significance:** 2
**Originality:** 2
**Overall Recommendation:** 4
**Confidence:** 3

**Summary:**

This paper proposes a framework to make the preference data selection method easier. They compare many methods under their framework and propose two sampling method.

**Compliance With Llm Reviewing Policy:**

Affirmed.

**Final Justification:**

The authors addressed several previously identified ambiguities during the rebuttal, which helps clarify the main concerns. Therefore, I maintain my weak accept score.

**Key Questions For Authors:**

Refer to above.

**Limitations:**

Refer to above.

**Strengths And Weaknesses:**

Strengths
1. The overall motivation is solid: how to select preference data to maximize the utility of available data resources is an important problem.
2. The paper presents a substantial number of experiments that are simple and easy to follow.
3. The comparison of various data sampling strategies is quite informative and offers useful references for future work.

Weaknesses
1. The sentence “considers prompts as contexts, and the system must select two ‘arms’ (responses) to annotate from a diverse pool of candidates” is central to your motivation but is not very easy to understand. It may help to include a brief, intuitive explanation of this idea near the results in Figure 1.
2. The methodology illustrated in Figure 2 is clearly presented and appears reasonable, and the final performance seems constrained by the data selection strategy. However, other researchers are likely to use a similar pipeline; it would be helpful to clarify what is unique about your pipeline compared with standard practice.
3. In Table 2, it would be good to clearly state whether larger numbers indicate better performance. It appears that your two methods are slightly worse on these metrics, and your proposed methods should be explicitly marked so that readers do not need to repeatedly look back to the text to identify them.
4. It seems that Figure 3 is mainly demonstrating the effectiveness of UltraFeedback, rather than directly highlighting the benefits of your two sampling strategies. If so, this distinction should be made clearer.
5. Overall, the paper is solid, but the main takeaways are not very prominent. The conclusions and your concrete recommendations could be further emphasized to make the contributions stand out more clearly.

---

> ### Author Rebuttal · Authors · 2026-03-31
>
> Dear Reviewer YpJu,
>
> Thank you for the time and effort in reviewing our submission and the helpful feedback you provided. We are happy to hear that you agree on the significance and importance of the problem we are tackling (selecting responses for preference data) and our extensive evaluations, along with their impact as a useful resource for future work.
>
> We want to address and clarify your concerns and questions below:
>
> 1. **Clarification of motivation sentence (Q1)**
>
>     Upon your feedback, we rephrased the motivation to: "Our framework is modeled after the contextual dueling bandit problem [Dudik et al., 2015]. In this setup, the prompt serves as the context, and the objective is to select two candidate responses (the arms) from a diverse pool for annotation." in order to improve its clarity. We hope this revision clarifies the connection to the bandit framework. Should any aspect remain unintuitive, please let us know, and we would be happy to provide further rephrasing.
>
> 2. **Differentiation from similar existing pipelines (Q2)**
>
>     Our ActiveUltraFeedback pipeline represents a novel configuration of methodologies, distinct from similar existing pipelines; the architectural framework and the UltraFeedback selection method is adapted from previous work [Cui et al., 2024, Lambert et al., 2025], some response pair selection methods have been proposed for the duelling bandit problem [Dwaracherla et al., 2024], and ENNs [Osband et al., 2023] are standard for uncertainty quantification. However, the resulting combination is unique, and we propose two novel response pair selection methods (DRTS and DeltaUCB). Our work constitutes a significant extension to these foundational efforts.
>
> 3. **"Table 2, clearly state whether larger numbers indicate better performance and mark proposed methods" (Q3)**
>
>     Thank you for noting it. We added arrows to clarify whether higher or lower means better performance and marked our novel methods with a dagger, similar to Figure 1.
>
> 4. **"Figure 3 is mainly demonstrating the effectiveness of UltraFeedback" (Q4)**
>
>     We respectfully disagree on this point. Figure 3 explicitly highlights the superior performance and sample efficiency of our novel selection strategies, and it refutes the premise that UltraFeedback selection is effective, especially for downstream training. We emphasize the critical distinction between the UltraFeedback dataset (the shared prompt pool) and the UltraFeedback selection method (the response pair selection strategy). Since all evaluated methods operate on the exact same underlying prompts, the performance gaps directly demonstrate how our novel strategies (DRTS and DeltaUCB) achieve superior performance with significantly fewer samples than the UltraFeedback selection baseline for downstream evaluations. We have made this distinction explicit by updating the caption for Figure 3 and revising the terminology throughout the main text.
>
> 5. **"Main takeaways are not very prominent" (Q5)**
>
>     As we discuss in the paper, our main takeaways are
>
>     - the ActiveUltraFeedback pipeline itself, which provides a modular platform for researchers and practitioners to streamline data collection and annotation;
>     - the introduction of novel acquisition functions, DRTS and DeltaUCB; and
>     - a significant reduction in required annotations, achieving up to 6x efficiency gains while maintaining or exceeding baseline performance. We further elaborate on this point in our response to Reviewer VmuH.
>
>     These core contributions are explicitly stated and emphasized in the Introduction, Abstract, and Conclusion sections.
>
> We believe these revisions address your concerns and substantially improve the clarity of the paper. We are happy to provide any further clarifications.

---

> > ### Author Rebuttal · Reviewer_YpJu · 2026-04-03
> >
> > Thank you for the detailed and thoughtful rebuttal. I appreciate the authors’ efforts in clarifying my concerns and providing additional explanations. I will maintain my initial score.

---

### Decision · Program_Chairs · 2026-04-30

**Decision:**

Accept (regular)

**Comment:**

This paper addresses an important problem in RLHF: how to generate informative preference data under limited annotation budgets. After reading the reviews, rebuttal, and follow-up discussion, I find the paper technically sound and sufficiently valuable for acceptance. Three reviewers remained positive after rebuttal, and the main strengths consistently mentioned were the broad empirical study across datasets and optimization algorithms, the focus on both reward-model and downstream policy performance, and the practical finding that the proposed delta-based selection strategies can recover most of the training signal with substantially fewer annotations than static baselines.

The rebuttal also meaningfully strengthened the paper. Concerns about computational cost, hyperparameter fairness, judge reliability, and sensitivity to the response pool were addressed with additional clarification and experiments, including cost tradeoff analysis, human-correlation evidence for the LLM judge, and analysis of model-pool diversity.

The main remaining concern is originality. One reviewer maintained a weak reject because they viewed the work as an incremental combination of existing ingredients rather than a fundamental algorithmic breakthrough. I agree that the novelty is more moderate than radical. However, I do not find this sufficient for rejection. Beyond introducing DRTS and DeltaUCB, the paper provides a careful large-scale study that yields useful empirical insights about preference data generation, especially the importance of quality deltas and the limited downstream utility of standard regret-minimizing dueling strategies in this setting. This kind of improved understanding is valuable and likely useful to the community.

Overall, I recommend acceptance. The paper would benefit from clearly positioning its contribution as a strong empirical and methodological study with moderate algorithmic novelty, but the current work is solid, well evaluated, and useful.